# Associations between multimorbidity and adverse health outcomes in UK Biobank and the SAIL Databank: A comparison of longitudinal cohort studies

Peter Hanlon[1], Bhautesh D. Jani[1], Barbara Nicholl[1], Jim Lewsey[2], David A. McAllister[3‡], Frances S. Mair[1‡*]

1 General Practice and Primary Care, Institute of Health and Wellbeing, University of Glasgow, Glasgow, United Kingdom, 2 Health Economics and Health Technology Assessment, Institute of Health and Wellbeing, University of Glasgow, Glasgow, United Kingdom, 3 Public Health, Institute of Health and Wellbeing, University of Glasgow, Glasgow, United Kingdom

‡ These authors are joint senior authors on this work.
* frances.mair@glasgow.ac.uk

**Data Availability Statement:** UK Biobank data can be obtained from UK Biobank project site, subject to successful registration and application process.

## Abstract

### Background

Cohorts such as UK Biobank are increasingly used to study multimorbidity; however, there are concerns that lack of representativeness may lead to biased results. This study aims to compare associations between multimorbidity and adverse health outcomes in UK Biobank and a nationally representative sample.

### Methods and findings

These are observational analyses of cohorts identified from linked routine healthcare data from UK Biobank participants ($n = 211,597$ from England, Scotland, and Wales with linked primary care data, age 40 to 70, mean age 56.5 years, 54.6% women, baseline assessment 2006 to 2010) and from the Secure Anonymised Information Linkage (SAIL) databank ($n = 852,055$ from Wales, age 40 to 70, mean age 54.2, 50.0% women, baseline January 2011). Multimorbidity ($n = 40$ long-term conditions [LTCs]) was identified from primary care Read codes and quantified using a simple count and a weighted score. Individual LTCs and LTC combinations were also assessed. Associations with all-cause mortality, unscheduled hospitalisation, and major adverse cardiovascular events (MACEs) were assessed using Weibull or negative binomial models adjusted for age, sex, and socioeconomic status, over 7.5 years follow-up for both datasets.

Multimorbidity was less common in UK Biobank than SAIL (26.9% and 33.0% with $\geq 2$ LTCs in UK Biobank and SAIL, respectively). This difference was attenuated, but persisted, after standardising by age, sex, and socioeconomic status. The association between increasing multimorbidity count and mortality, hospitalisation, and MACE was similar between both datasets at LTC counts of $\leq 3$; however, above this level, UK Biobank under-estimated the risk associated with multimorbidity (e.g., mortality hazard ratio for 2 LTCs

Further details can be found at https://www.ukbiobank.ac.uk/. SAIL data are available upon application to the SAIL Information Governance Review Panel. Further details can be found at https://saildatabank.com/application-process/. All syntax underlying the analysis presented will be returned to UK Biobank for record, along with all model outputs from SAIL, as per the UK Biobank material transfer agreement. This syntax will be available to third party researchers from UK Biobank.

**Funding:** PH is funded through a Clinical Research Training Fellowship from the Medical Research Council (Grant reference: MR/S021949/1). DM is funded via an Intermediate Clinical Fellowship and Beit Fellowship from the Wellcome Trust - 201492/Z/16/Z. The funders had no role in study design, data collection and analysis, decision to publish, or preparation of the manuscript.

**Competing interests:** I have read the journal's policy and the authors of this manuscript have the following competing interests: FM is principal supervisor of PH (first author) who is funded by a MRC Clinical Research Training Fellowship (Grant reference: MR/S021949/1) which supported PH to do this work. FM is also Principle Investigator or Co-Investigator on grants funded by the MRC, NIHR, Wellcome, CSO, and EPSRC to undertake multimorbidity research. The funds go to FM's institution, the University of Glasgow.

**Abbreviations:** AIC, Akaike information criterion; LTC, long-term condition; MACE, major adverse cardiovascular event; SAIL, Secure Anonymised Information Linkage.

1.62 (95% confidence interval 1.57 to 1.68) in SAIL and 1.51 (1.43 to 1.59) in UK Biobank, hazard ratio for 5 LTCs was 3.46 (3.31 to 3.61) in SAIL and 2.88 (2.63 to 3.15) in UK Biobank). Absolute risk of mortality, hospitalisation, and MACE, at all levels of multimorbidity, was lower in UK Biobank than SAIL (adjusting for age, sex, and socioeconomic status). Both cohorts produced similar hazard ratios for some LTCs (e.g., hypertension and coronary heart disease), but UK Biobank underestimated the risk for others (e.g., alcohol-related disorders or mental health conditions). Hazard ratios for some LTC combinations were similar between the cohorts (e.g., cardiovascular conditions); however, UK Biobank underestimated the risk for combinations including other conditions (e.g., mental health conditions). The main limitations are that SAIL databank represents only part of the UK (Wales only) and that in both cohorts we lacked data on severity of the LTCs included.

## Conclusions

In this study, we observed that UK Biobank accurately estimates relative risk of mortality, unscheduled hospitalisation, and MACE associated with LTC counts ≤3. However, for counts ≥4, and for some LTC combinations, estimates of magnitude of association from UK Biobank are likely to be conservative. Researchers should be mindful of these limitations of UK Biobank when conducting and interpreting analyses of multimorbidity. Nonetheless, the richness of data available in UK Biobank does offers opportunities to better understand multimorbidity, particularly where complementary data sources less susceptible to selection bias can be used to inform and qualify analyses of UK Biobank.

## Author summary

### Why was this study done?

- Multimorbidity, the presence of multiple long-term conditions (LTCs), is associated with a range of adverse health outcomes.

- The UK Biobank cohort study has gathered and linked genetic, physical, and clinical information on a population scale providing unique opportunities to study the impact of multimorbidity.

- However, participants in UK Biobank appear on average to be healthier than the general population ("healthy volunteer bias") and it is not clear if this selection bias affects estimates of the impact of multimorbidity using UK Biobank.

### What did the researchers do and find?

- We compared the prevalence of multimorbidity, and the impact of multimorbidity on adverse health outcomes, in UK Biobank and in a representative sample of people from Wales, UK (SAIL databank).

- While multimorbidity was less common in UK Biobank, the relationship between number of LTCs and mortality, hospital admissions, and major adverse cardiovascular events (MACEs) was similar between UK Biobank and SAIL at lower levels of

multimorbidity (e.g., 2 or 3 LTCs) and for many common LTCs (e.g., hypertension, coronary artery disease, and chronic obstructive pulmonary disease).

- However, for people with higher LTC counts (e.g., 4 or more), or with specific LTCs such as mental health conditions, UK Biobank underestimates the risk of mortality, hospitalisation, and MACEs.

## What do these findings mean?

- The wide range of measures gathered by UK Biobank make it a valuable resource for studying multimorbidity, and our study suggests that analyses of modest levels of multimorbidity (such as people with 2 or 3 LTCs) are likely to yield reliable estimates.

- However, for people with a higher number of LTCs or with LTCs such as mental health conditions, alcohol-related disorders, or addiction, estimates based on UK Biobank data are likely to be conservative compared to a representative sample.

- Ideally, future LTC and multimorbidity research should combine insights from both representative routine data and information rich research cohorts such as UK Biobank.

- These analyses are limited by a lack of data on the severity of LTCs.

## Introduction

Multimorbidity, the coexistence of multiple long-term conditions (LTCs) within an individual, is a global clinical and public health priority [1]. Multimorbidity is common [2], increasing in prevalence [3], and associated with adverse health outcomes [4,5]. However, important questions remain about how best to identify, quantify, prevent, and manage multimorbidity within modern healthcare. There is also growing interest in the implications of specific combinations, or clusters, of LTCs, and the identification of specific targets for intervention [6,7]. An important resource for addressing such questions are large cohort studies with linkage to routine healthcare data. One such resource is the UK Biobank.

UK Biobank is a large cohort study of approximately 500,000 people recruited from across England, Scotland, and Wales. Strengths of UK Biobank include its size, detailed baseline assessment and measurements including biological samples, follow-up assessments for certain issues, and the availability of long-term linkage to outcome data. This has allowed investigators to explore various aspects of multimorbidity including demographic patterns [4], prevalence of disease clusters [8], association with related states such as frailty or sarcopenia [9,10], the impact of lifestyle factors in the context of multimorbidity [11], and associations between multimorbidity and adverse health outcomes [4,12–15]. Many such analyses would not have been possible using routine data alone, and the sample size of UK Biobank provides considerable statistical power. However, one major limitation of UK Biobank is that its response rate to recruitment invitation, approximately 5%, is substantially lower than many cohort studies [16,17]. UK Biobank participants are recognised to be less socioeconomically deprived, have fewer lifestyle risk factors, and have a lower prevalence of LTCs than the UK population in general [18]. This presents challenges for studying multimorbidity, in particular in relation to the risk of selection bias [19,20].

It has been claimed that while UK Biobank cannot be described as representative, associations between risk factors and outcomes are nonetheless reliable [21]. Others, however, have questioned this assertion [22]. In particular, it is argued that notwithstanding the large sample size and presence of participants with multiple LTCs, selection pressures within the sample may lead to collider bias, resulting in biased associations between risk factors and outcomes. A collider is a variable that is causally influenced by 2 other variables [19]. For example, if people with multiple LTCs and people who are more likely to be hospitalised or die are less likely to be recruited into UK Biobank (i.e., a "healthy cohort" effect), this could bias the association between LTCs and those outcomes observed in UK Biobank.

A recent study compared associations between a range of risk factors and mortality in UK Biobank and in national surveys with considerably higher response rates and reported that the relative associations between risk factors and outcomes were very broadly similar between the 2 sources [21]. It is not clear, however, if the same is true of multimorbidity, as the previous study did not assess LTCs nor did it explore combinations of risk factors. Furthermore, when studying multimorbidity, absolute risks of outcomes are important as well as relative associations. This previous comparison is also limited by the fact that although the health surveys used for comparison had higher response rates than UK Biobank, they still relied on voluntary recruitment and are therefore also susceptible to selection bias. This study seeks to compare the relationship between multimorbidity and adverse health outcomes in UK Biobank with a representative, unselected population identified from routine health records. This is important as it addresses the question of whether UK Biobank analyses can produce reliable estimates of the associations between multimorbidity and adverse health outcomes. Understanding and quantifying the reliability of UK Biobank in this context is needed if the strengths mentioned above are to be harnessed to meet the rising challenge of multimorbidity.

Our study aims (i) to compare the prevalence of multimorbidity in UK Biobank with an unselected population identified from the Secure Anonymised Information Linkage (SAIL) databank; and (ii) to compare the associations between multimorbidity and all-cause mortality, unscheduled hospitalisation, and major adverse cardiovascular events (MACEs) between the datasets, in both relative and absolute terms.

## Methods

This is an observational analysis of multimorbidity in cohorts identified from UK Biobank and the SAIL databank. Unless otherwise stated in this section, all analyses were conducted according to a prespecified analysis plan (S1 Protocol). This study is reported as per the Strengthening the Reporting of Observational Studies in Epidemiology (STROBE) guideline (S1 Checklist).

### Data sources

**UK Biobank.**  UK Biobank participants were recruited between 2006 and 2010 by postal invitation (5% response rate). Eligible participants had to be registered with a General Practitioner, aged between 40 and 70, and live within 20 miles of one of 22 assessment centres in England, Scotland, and Wales. Participants answered a touchscreen questionnaire, a nurse interview, and had physical measurements taken. Participants also consented to data linkage to healthcare records including primary care, hospital episode statistics, and mortality registers.

At present, data on linked primary care data are only available for a subset of participants. The availability of primary care data depended on the electronic medical record system used by the practice (data only currently available from certain systems) rather than any

participant-level factors. This subset is representative of the UK Biobank cohort in terms of age, sex, socioeconomic status, lifestyle factors, and self-reported LTCs (S1 Table). To allow a consistent approach to identifying multimorbidity between UK Biobank and SAIL, we restricted our analysis to this subset of participants with primary care data. We further excluded participants from practices using the "Vision" primary care system, as data were not available from these practices for participants who had died since baseline assessment.

This analysis was performed as part of UK Biobank project 14151.

**SAIL.** Data on our unselected community comparison were taken from the SAIL databank [23]. SAIL collected linked primary care, hospital, and mortality data from participating general practices within Wales. Participating practices cover approximately 70% of the population of Wales and are representative of the wider population in terms of age, sex, and socioeconomic deprivation [24,25]. Comparison between the SAIL population and the overall population of Wales are reported elsewhere [25].

From this population, we identified all participants registered with a participating practice on 1 January 2011, and who were aged between 40 and 70 years (to mirror the age criteria of UK Biobank). This time was chosen as electronically coded data is most complete from this point and is similar to the timing of recruitment to UK Biobank.

## Assessment of multimorbidity

We quantified multimorbidity using a count of 40 LTCs originally developed for a large epidemiological study in Scotland, identified using Read codes [26]. Read codes are used in primary care to record diagnoses as well as coding events such as prescriptions. The coding system is common to Wales and the rest of the UK. LTCs were identified from Read codes occurring prior to the baseline assessment, using the criteria described by Barnett and colleagues [26]. For most conditions, any diagnostic Read code occurring prior to baseline assessment was taken to indicate the presence of the LTC (as per the previous study on which the definitions were based) [26]. For some prespecified conditions, the presence of a recent diagnostic code (e.g., depression, anxiety) or of specific prescriptions dispensed in the year prior to baseline (e.g., asthma, constipation, pain, dyspepsia, migraine, epilepsy) were used to confirm if the condition were current (also according to definitions described by Barnett and colleagues) [26]. A list of all Read codes used for diagnoses and prescriptions is included in the Supporting information (S1 and S2 Files).

At the time of baseline assessment, many LTCs in the UK were managed and coded according to the Quality Outcomes Framework. These standards were applied in Wales similarly to the rest of the UK. Median time from practice registration to baseline assessment (from which diagnostic Read codes were assessed) was 14.6 years (interquartile range 7.4 to 23.3) in UK Biobank and 14.8 years (interquartile range 10.2 to 18.8) in SAIL.

For both UK Biobank and SAIL, we calculated the count of LTCs for each individual at baseline, which was treated as a numerical variable.

In order to enhance our understanding of the influence of type as well as number of LTCs, we also calculated the Cambridge multimorbidity score [5]. The Cambridge multimorbidity score has been developed based on this same list of 40 LTCs but is a weighted score used to predict mortality, hospitalisation, and primary care use. Weights are applied to each LTC and then summed to give an overall score. Separate weights are available for each outcome, as well as a generic "overall" weight. For this study, we used weights specific to each outcome where available (i.e., for mortality and hospitalisation). Using the Cambridge multimorbidity score allows us to assess the impact of type of LTC, in addition to a simple count, when looking at associations with hospitalisation and mortality using baseline data for each participant in each dataset.

We also analysed each of the 40 LTCs separately, assessing the relationship between each LTC and all-cause mortality. Finally, we identified combinations of 3 LTCs from previous studies of common co-occurring LTCs (clusters) [7,12]. We assessed the relationship between all possible combinations of the conditions within each cluster and all-cause mortality, unscheduled hospitalisation, and MACE. For example, for the cluster of coronary heart disease, diabetes, and stroke, we assessed the risk of adverse outcomes in each condition individually, in all pairwise combinations, and in people with all 3 conditions. The clusters of conditions assessed were as follows: coronary heart disease, diabetes, and stroke or transient ischaemic attack; hypertension, diabetes, and pain; pain, deafness, and irritable bowel syndrome; depression, pain, and anxiety; asthma, pain, and chronic obstructive pulmonary disease; alcohol-related disorders, illicit drug use, and pain; coronary heart disease, pain, and depression.

## Covariates

Age was taken as age at baseline for both datasets (this corresponded to date of recruitment for UK Biobank, and 1 January 2011 for SAIL participants) and was treated as a numerical variable. Sex was coded as male or female and treated as categorical. Socioeconomic deprivation was assessed using Townsend scores: an area-based measure of socioeconomic deprivation [27]. Baseline UK Biobank data are linked to Townsend scores based on participant postcodes linked to previous (2001) census data. To allow comparison to a similar measure of socioeconomic deprivation in SAIL, we obtained Townsend scores based on the same census data (2001) for Local Super Output Areas in Wales, and linked these to baseline SAIL data. Townsend scores were analysed as a numerical variable.

## Outcomes

Outcomes of interest were all-cause mortality, unscheduled (urgent or emergency) hospital admission, and MACEs. For both datasets, outcomes were assessed by linkage to Office for National Statistics mortality registers and to hospital inpatient data (Hospital Episode Statistics or Patient Episode Database for Wales), respectively. Unscheduled hospital admission was defined as any inpatient hospital episode coded as "urgent" or "emergency" (excluding admissions coded as "elective"). MACE was defined as hospital admission with myocardial infarction (ICD-10 code I21), stroke (I63 or I64), or cardiovascular death (ICD-10 code for underlying cause of death beginning with "I"). In SAIL, linked outcome data were available for 7.5 years follow-up (censored at 1 July 2018). Participants were censored at date of death, deregistration from a participating practice, or 7.5 years follow-up, whichever occurred first. For UK Biobank, to ensure comparable follow-up periods for outcome assessment, we censored participants similarly at death, deregistration, or 7.5 years.

## Statistical analysis

SAIL data are stored and analysed within a secure repository allowing remote access. Export is limited to aggregate data. Therefore, comparison between data sources was achieved by fitting models within SAIL and then exporting model parameters from which the relationships between covariates could be summarised and presented. This process is described in detail for each comparison below. This approach was chosen because outputs from regression models are considered to be a lower risk to privacy than are frequency tables while still allowing flexibility in subsequent reporting and comparisons.

**Comparison of multimorbidity distribution.** We first plotted the distribution of LTC counts in each dataset and compared these graphically to assess the observed distribution of multimorbidity in each dataset.

We then compared the distribution of LTC count in UK Biobank with SAIL after standardising SAIL to the age/sex distribution, and then the age/sex/socioeconomic status distribution, of UK Biobank using a form of standardisation analogous to indirect standardisation. To do this, we first modelled the distribution of LTC count in SAIL on age and sex, and on age, sex, and Townsend score. We assessed Poisson and negative binomial models initially. We used up to 2 fractional polynomial terms (from the set −2, −1, −0.5, 0, 0.5, 1, 2, and 3 recommended by Royston and colleagues) to model nonlinear relationships between age and Townsend score and LTC counts [28]. The best-fitting fractional polynomials were identified using backward selection in the *mfp* package in R. We also explored interaction terms between each of the explanatory variables. Candidate models were assessed using the Akaike information criterion (AIC), by comparing the likelihood ratio and assessing the dispersion statistic for each model. We also compared the best-fitting models to zero-inflated models using Vuong tests. Finally, we plotted expected and observed LTC counts within strata of age, sex, and Townend score. The negative binomial model was found to fit the data well. We exported the model coefficients, variance–covariance matrix, and dispersion parameter from the secure platform, along with diagnostic plots summarising model fit (S1 Fig). This allowed us to model the expected distribution of LTC counts in SAIL conditional on age/sex and on age/sex/Townsend scores. Participants with missing data for Townsend scores were excluded from these models.

Next, we calculated the expected LTC count in UK Biobank based on the models fitted in the SAIL data. To do this, we aggregated the UK Biobank dataset by age (1-year bands), sex, and Townsend score (0.1-point intervals). We summed the total number of participants, as well as the number of participants with each LTC count, in each stratum. We used the coefficients of the negative binomial models exported from SAIL to calculate the expected mean LTC count for each stratum. Next, we used the cumulative density function for the negative binomial distribution to calculate the expected proportion of participants with each LTC count within each stratum. Finally, each expected proportion was multiplied by the number of UK Biobank participants in each stratum to obtain the expected number of participants with each LTC count. We then summed these expected values across strata to calculate the expected number of UK Biobank participants with each LTC count.

To calculate 95% confidence intervals for these estimates, we obtained 10,000 samples of the intercept, age, sex, and Townsend score coefficients by sampling from a multivariate normal distribution where the parameters were the point estimates and variance–covariance matrix for the SAIL regression models. We then repeated the above process with each of the 10,000 samples to obtain 10,000 estimates of the expected count, which we then summarised by the mean, 2.5th and 97.5th centiles to obtain 95% confidence intervals.

**Modelling all-cause mortality and MACE.** For all-cause mortality and MACE, we used parametric survival models to assess the relationship between LTC counts and each outcome. This allowed us to export model summary statistics from the SAIL secure platform, which could facilitate estimation of predicted event rates for any combination of covariate levels. Before fitting models, we plotted log(time) against log(−log(Kaplan–Meier estimates)) for strata of each covariate. We found that for age (5-year bands), sex, and Townsend scores (1-point intervals), lines were linear and parallel, suggesting that Weibull models were an adequate fit for the data. Up to 2 fractional polynomial terms (selected using similar methods to the previous section) and interaction terms were explored. Best-fitting models were selected by comparing AIC and likelihood ratios. We finally compared Weibull models to more flexible Generalised Gamma models. Weibull models including fractional polynomials for age and Townsend scores, as well as interaction terms between LTC count and age, and LTC count and sex, were found to fit the data best. We summarised the overall fit by plotting predicted and observed 5-year event rates (S2 and S3 Figs). Separate models were fitted for UK Biobank

and SAIL. For MACE, we assessed time to first event and used cause-specific models (censoring participants dying of noncardiovascular causes and coding as event-status "0"). These models were then used to obtain predicted 5-year mortality at different values of age, sex, Townsend score, and LTC count. This allowed comparison of the absolute risk of mortality for different LTC counts between UK Biobank and SAIL. We then fit a further model treating LTC count as categorical (0, 1, 2, 3, 4, 5, and more than 5) and obtained hazard ratios to assess any difference in relative risk of mortality.

We repeated this process using the mortality weightings for the Cambridge multimorbidity score, in place of the LTC count.

**Modelling unscheduled hospitalisations.**   For unscheduled hospitalisation, we fit count-based models using a similar strategy to that described above for modelling number of LTCs. Briefly, we compared Poisson and negative binomial models (with and without fractional polynomials and interaction terms) using likelihood ratios and dispersion statistics. We also assessed for zero inflation using Vuong tests. Overall fit of the final model was assessed by plotting observed and expected numbers of hospitalisations for each cohort as a whole, and within strata of age, sex, and socioeconomic status. Negative binomial models were found to best fit the data and included nonlinear terms for LTC count and age, as well as interactions between age and sex; age and LTC count; and sex and LTC count. Model fit is summarised in S4 and S5 Figs. Models for UK Biobank and SAIL, respectively, were used to assess hospitalisation rates per 1,000 person years at given levels of LTC count (plus age, sex, and socioeconomic status). We then repeated this process using the hospitalisation weights from the Cambridge multimorbidity score.

**Modelling specific LTCs.**   We used Weibull models to assess the relationship between each individual LTC and mortality, adjusting for age, sex, and socioeconomic status.

Finally, we assessed the relationship between prevalent combinations of 3 LTCs and each outcome (all-cause mortality, unscheduled hospitalisation, and MACE) using Weibull or negative binomial models adjusted for age, sex, and socioeconomic status.

**Sensitivity analyses.**   In sensitivity analysis, we repeated analyses of the association between LTC count and mortality, hospitalisations, and MACE restricting the UK Biobank cohort to those participants who attended one of 2 assessment centres based in Wales. This was to assess if differences between UK Biobank and the community cohort were driven by differences between Wales and the rest of the UK. Separately, we also repeated the SAIL analyses restricting SAIL to participants registered in Cardiff and Swansea (the locations of the UK Biobank assessment centres in Wales). These analyses were not prespecified, being conducted following initial peer review.

## Ethical approval

The UK Biobank has full ethical approval from the NHS National Research Ethics Service (16/NW/0274). All participants gave informed consent for participation in UK Biobank. Permission to access and analyse UK Biobank data was approved under UK Biobank project 14151. SAIL analyses were approved by SAIL Information Governance Review Panel (Project 0830).

## Results

Primary care data were available for 211,597 UK Biobank participants and were compared to 852,349 SAIL participants. Baseline characteristics are shown in Table 1. UK Biobank had a greater proportion of female participants. UK Biobank participants also tended to be more affluent (42.0% in the most affluent quintile of the UK as a whole), whereas SAIL participants were on average less affluent than the UK population as a whole (e.g., 10.6% in the most

**Table 1. Baseline demographic characteristics.**

|  | SAIL | UK Biobank |
|---|---|---|
| Total | 852,349 | 211,597 |
| **Age** | | |
| Mean (SD) | 54.2 (8.6) | 56.5 (8.1) |
| **Sex** | | |
| Men | 426,372 (50.0%) | 96,060 (45.4%) |
| Women | 425,977 (50.0%) | 115,537 (54.6%) |
| Socioeconomic status* | | |
| Quintile 1 (most affluent) | 89,944 (10.6%) | 88,883 (42.0%) |
| Quintile 2 | 96,622 (11.3%) | 29,454 (13.9%) |
| Quintile 3 | 168,868 (19.8%) | 27,937 (13.2%) |
| Quintile 4 | 308,080 (36.1%) | 33,324 (15.7%) |
| Quintile 5 (most deprived) | 188,835 (22.2%) | 31,675 (15.0%) |
| Missing | - | 324 |
| **LTC count** | | |
| 0 | 281,580 (33.0%) | 102,547 (48.5%) |
| 1 | 196,584 (23.1%) | 51,955 (24.6%) |
| 2 | 151,043 (17.7%) | 29,149 (13.8%) |
| 3 | 97,946 (11.5%) | 14,924 (7.1%) |
| 4 | 58,672 (6.8%) | 7,156 (3.4%) |
| 5 | 33,130 (3.9%) | 3,353 (1.6%) |
| ≥6 | 33,394 (3.9%) | 2,403 (1.1%) |

*Quintiles of socioeconomic status (Townsend scores) are based on quintiles for the general UK population.

LTC, long-term condition; SAIL, Secure Anonymised Information Linkage; SD, standard deviation.

affluent quintile). UK Biobank participants with primary care data were similar to the wider UK Biobank cohort in terms of age, sex, socioeconomic status, and self-reported LTCs (S1 Table). Median follow-up time was 7.5 years in both datasets.

## Distribution of multimorbidity

The distribution of LTC counts for SAIL and UK Biobank are shown in Fig 1. Median LTC count was similar in SAIL (median 1 LTC, interquartile range 0 to 3) and UK Biobank (median 1 LTC, interquartile range 0 to 2); however, the distribution in SAIL was more skewed to the right. A far greater proportion of UK Biobank participants had no LTCs compared to SAIL. Few UK Biobank participants (2,403/211,597, 1.1%) had LTC counts greater than 5, whereas 3.8% (33,100/852,349) of SAIL participants had more than 5 LTCs.

Standardising SAIL to the age and sex distribution of UK Biobank had little impact on the difference in LTC counts (blue triangles on Fig 2)—as would be expected since the SAIL comparison was selected to have the same age range as UK Biobank. After additionally standardising by socioeconomic deprivation (red circles), however, the expected LTC counts in SAIL were closer to the observed counts in UK Biobank. These findings demonstrate that differences in socioeconomic deprivation account for a substantial proportion of the difference between UK Biobank and SAIL. Notably, however, there were still a greater proportion of UK Biobank participants with no LTCs.

## Association between multimorbidity and all-cause mortality

The association between LTC count and all-cause mortality in UK Biobank and SAIL, adjusted for age, sex, and socioeconomic deprivation, is summarised in Figs 3 and 4. The predicted absolute risk is shown in Fig 3 and demonstrates that, at all levels of LTC count, the adjusted

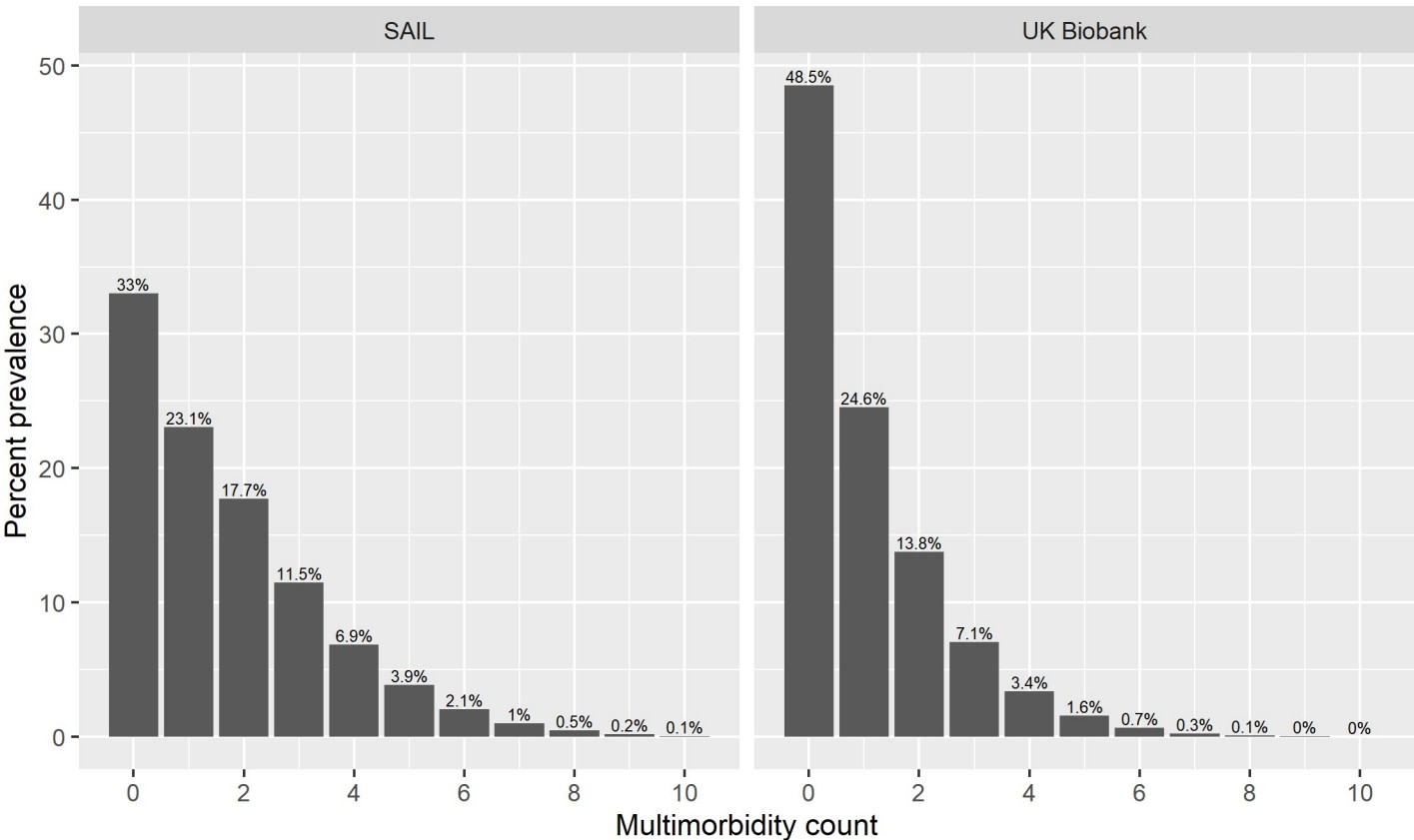

**Fig 1. Distribution of counts in UK Biobank and SAIL.** Bar plot showing the percentage of participants at baseline with each LTC count for UK Biobank and SAIL, respectively. LTC, long-term condition; SAIL, Secure Anonymised Information Linkage.

mortality rate is lower in UK Biobank than in SAIL (e.g., for a 60-year-old man with 2 LTCs, predicted 5-year mortality was 2.7% (95% CI 2.6 to 2.8) in UK Biobank and 4.6% (95% CI 4.4 to 4.8) in SAIL). This pattern is consistent across all levels of socioeconomic deprivation (S6 Fig). When using the mortality weightings from the Cambridge multimorbidity score, and thus considering the type as well as number of LTCs, the difference in absolute mortality risk between SAIL and UK Biobank persisted (e.g., for a 60-year-old man with a Cambridge mortality score of 12 [mean value in UK Biobank, approximately equivalent to a diagnosis of alcohol-related disorders: score 12.72, or the combination of diabetes, coronary heart disease, and hypertension: combined score 12.36], predicted 5-year mortality was 2.2% (95% CI 2.1 to 2.3) in UK Biobank and 4.0% (95% CI 4.0 to 4.1) in SAIL; see S7 Fig).

Fig 4 shows adjusted hazard ratios for mortality associated with the different LTC counts for UK Biobank and SAIL. As LTC counts increase, the difference in the relative association between UK Biobank and SAIL increases, with SAIL showing higher hazard ratios at greater LTC counts.

## Association between multimorbidity and unscheduled hospitalisation

The absolute and relative associations between LTC counts and unscheduled hospital admission are shown in Figs 5 and 6, respectively. Up to an LTC count of 3, the LTC count-specific rate of hospital admissions was similar between UK Biobank and SAIL. Above this level, the rate was higher in SAIL than in UK Biobank. At all levels of LTC count, the incident rate ratio

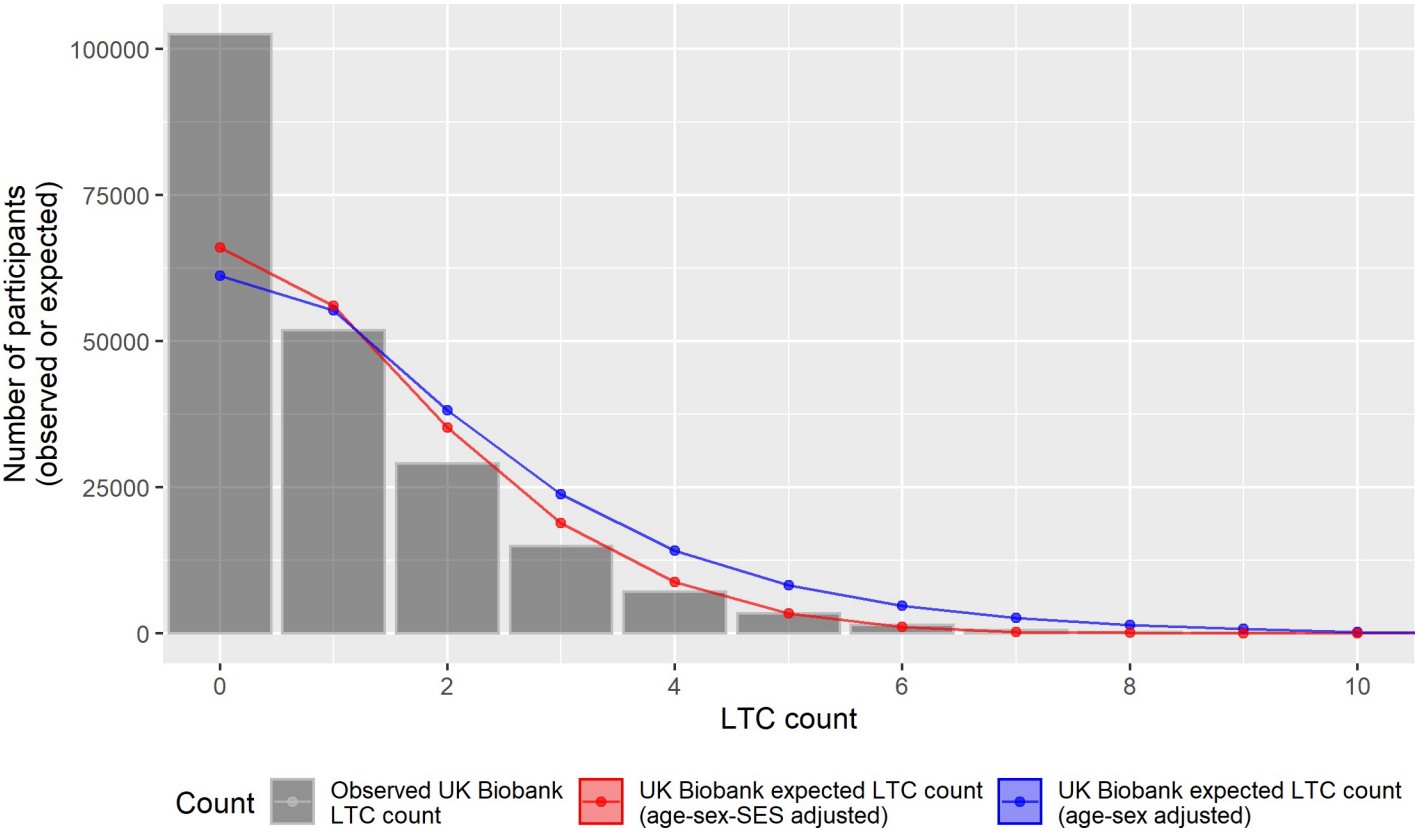

**Fig 2. Observed and expected LTC counts in UK Biobank.** The grey bars show the observed LTC counts in UK Biobank. The blue line shows the expected LTC counts in UK Biobank based on the age and sex adjusted regression models fitted in SAIL. The red line shows the expected LTC counts in UK Biobank based on age, sex, and socioeconomic status adjusted regression models fitted in SAIL. Observed and expected counts, with 95% confidence intervals, are also shown in S2 Table. LTC, long-term condition; SAIL, Secure Anonymised Information Linkage; SES, socioeconomic status.

for SAIL was higher than for UK Biobank. Also, as LTC count increases, the difference in relative association between UKB and SAIL increases. As with mortality, the pattern was similar when using the weighted Cambridge multimorbidity score, with similar hospitalisation rates at low levels but higher rates in SAIL at higher levels.

## Association between multimorbidity and MACE

Findings for MACE are shown in Fig 7 (for absolute risk) and Fig 8 (for relative associations). As for mortality and hospitalisation, absolute risk was lower in UK Biobank; however, the relative association was similar at lower LTC counts (below 5).

## Relationship between individual LTCs and all-cause mortality

The prevalence and hazard ratios for all-cause mortality for each individual LTC are shown in Fig 9. Results are adjusted for age, sex, and socioeconomic deprivation. Most LTCs were more prevalent in SAIL than in UK Biobank; however, the magnitude of difference was highly variable. For example, chronic kidney disease was more than 6 times more prevalent in SAIL than in UK Biobank, and dementia and illicit drug use were more than 5 times more prevalent. Several common conditions, such as coronary heart disease, hypertension, chronic obstructive pulmonary disease, and asthma, were all more than twice as prevalent in SAIL than in UK

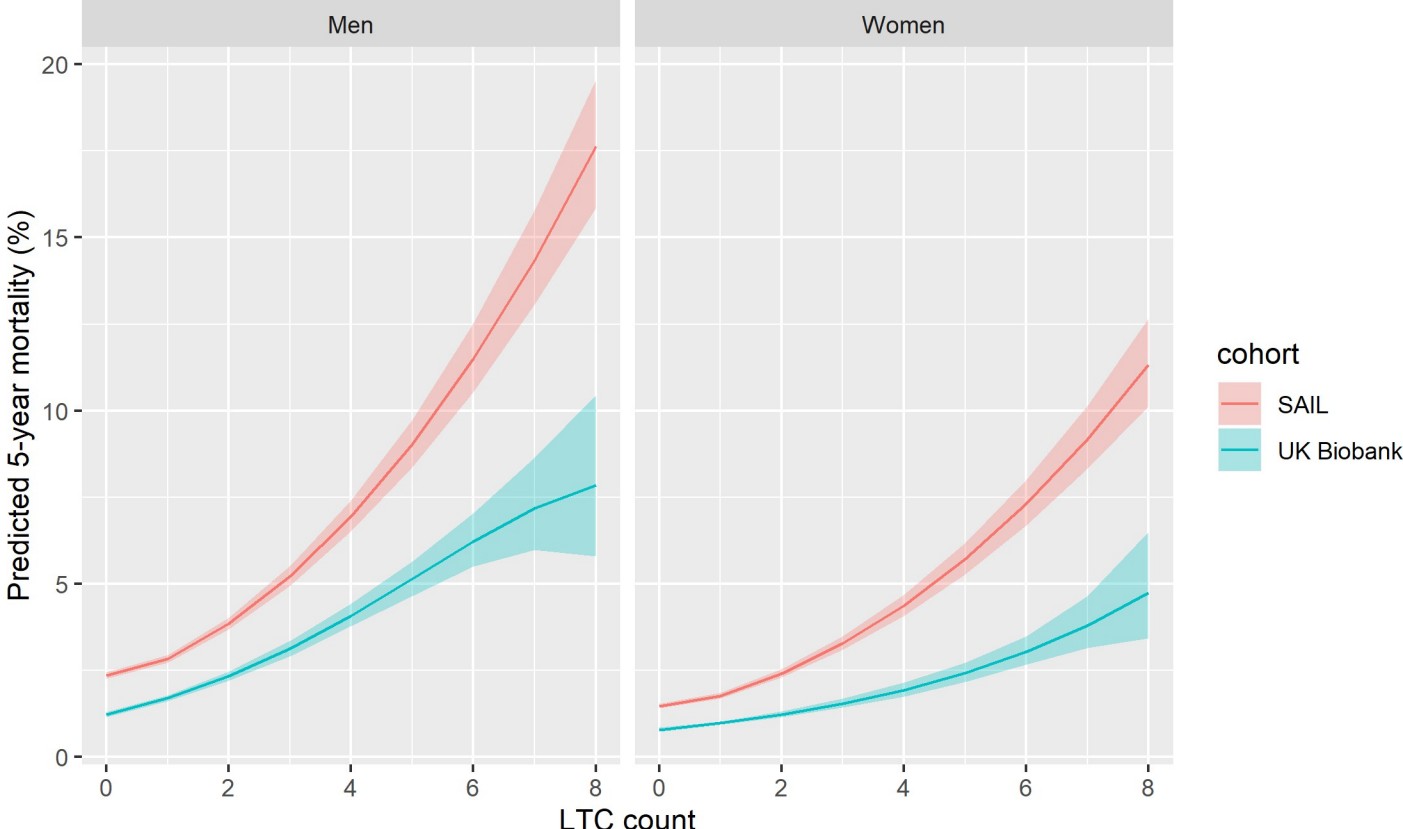

**Fig 3. Relationship between LTC count and absolute mortality risk for UK Biobank and SAIL.** Lines indicate the modelled 5-year mortality rate. Shaded area indicates 95% confidence interval. Predicted values are modelled at mean age for UK Biobank (56.5 years) and UK national mean Townsend score for socioeconomic status. LTC, long-term condition; SAIL, Secure Anonymised Information Linkage.

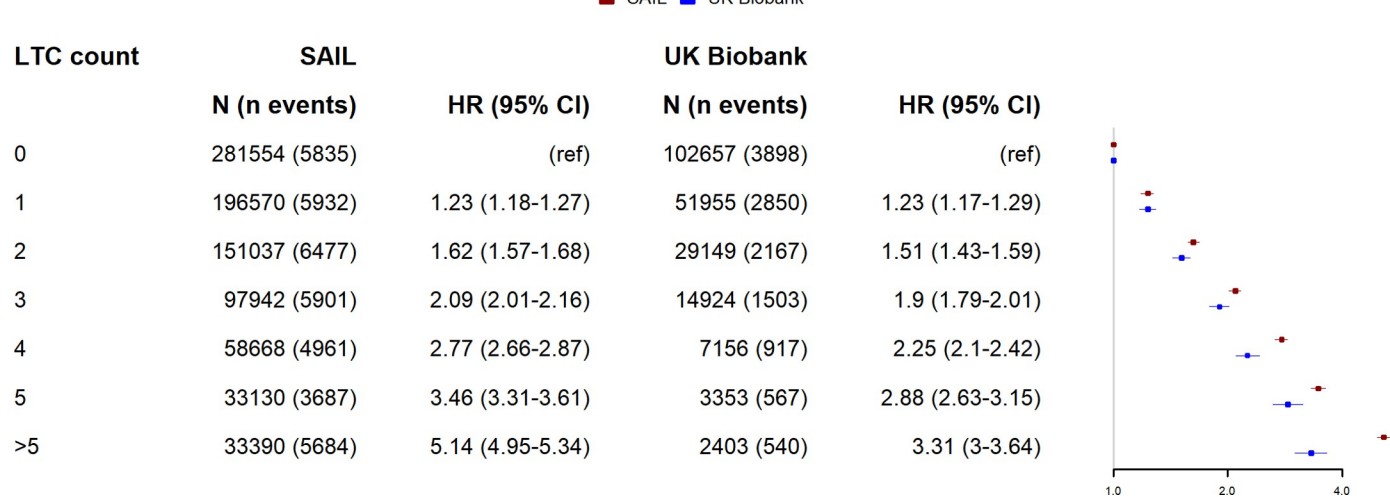

| LTC count | SAIL | | UK Biobank | | |
|---|---|---|---|---|---|
| | N (n events) | HR (95% CI) | N (n events) | HR (95% CI) | |
| 0 | 281554 (5835) | (ref) | 102657 (3898) | (ref) | |
| 1 | 196570 (5932) | 1.23 (1.18-1.27) | 51955 (2850) | 1.23 (1.17-1.29) | |
| 2 | 151037 (6477) | 1.62 (1.57-1.68) | 29149 (2167) | 1.51 (1.43-1.59) | |
| 3 | 97942 (5901) | 2.09 (2.01-2.16) | 14924 (1503) | 1.9 (1.79-2.01) | |
| 4 | 58668 (4961) | 2.77 (2.66-2.87) | 7156 (917) | 2.25 (2.1-2.42) | |
| 5 | 33130 (3687) | 3.46 (3.31-3.61) | 3353 (567) | 2.88 (2.63-3.15) | |
| >5 | 33390 (5684) | 5.14 (4.95-5.34) | 2403 (540) | 3.31 (3-3.64) | |

**Fig 4. Relationship between LTC count and hazard ratio for mortality in UK Biobank and SAIL.** Points indicate hazard ratios adjusted for age, sex, and socioeconomic status. Whiskers indicate 95% CIs. N = total number of participants per LTC count, n events = number of events. CI, confidence interval; HR, hazard ratio; LTC, long-term condition; SAIL, Secure Anonymised Information Linkage.

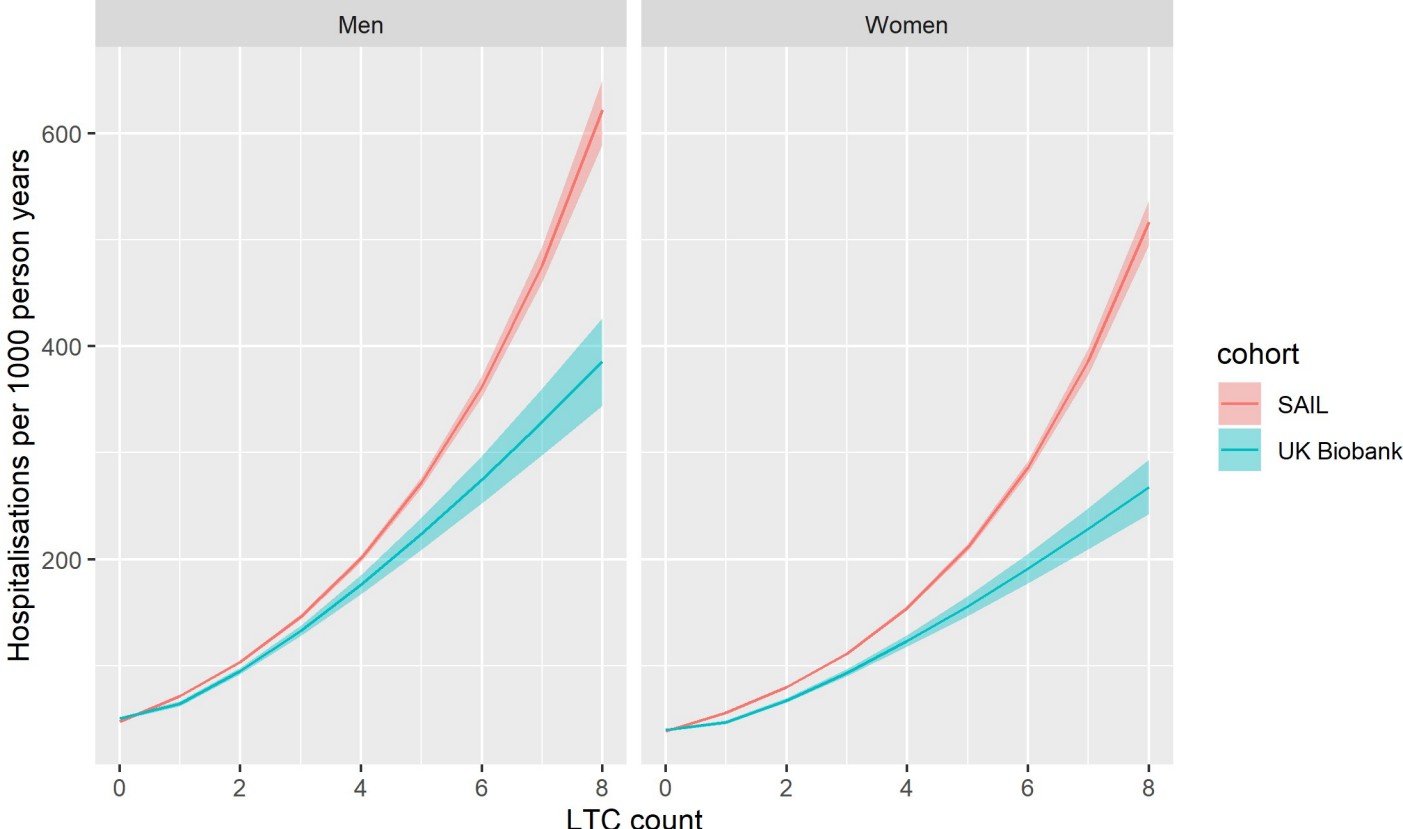

**Fig 5. Relationship between LTC count and absolute risk of unscheduled hospitalisation for UK Biobank and SAIL.** Lines indicate the modelled rate of unscheduled hospitalisations per 1,000 person years observation. Shaded area indicates 95% confidence interval. Predicted values are modelled at mean age for UK Biobank (56.5 years) and UK national mean Townsend score for socioeconomic status. LTC, long-term condition; SAIL, Secure Anonymised Information Linkage.

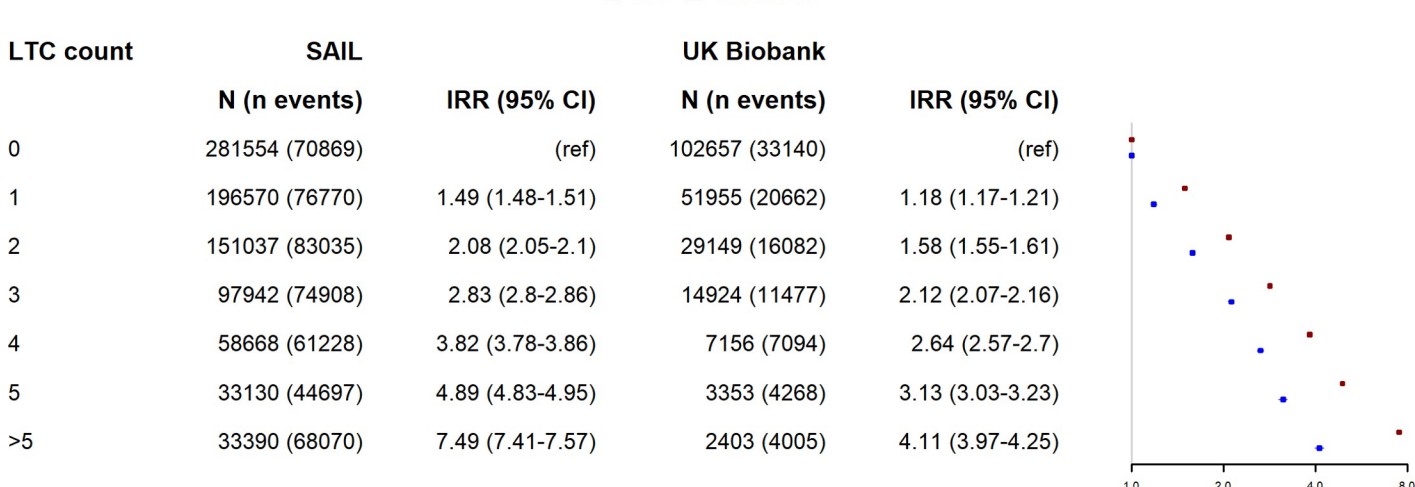

| LTC count | SAIL N (n events) | SAIL IRR (95% CI) | UK Biobank N (n events) | UK Biobank IRR (95% CI) |
|---|---|---|---|---|
| 0 | 281554 (70869) | (ref) | 102657 (33140) | (ref) |
| 1 | 196570 (76770) | 1.49 (1.48-1.51) | 51955 (20662) | 1.18 (1.17-1.21) |
| 2 | 151037 (83035) | 2.08 (2.05-2.1) | 29149 (16082) | 1.58 (1.55-1.61) |
| 3 | 97942 (74908) | 2.83 (2.8-2.86) | 14924 (11477) | 2.12 (2.07-2.16) |
| 4 | 58668 (61228) | 3.82 (3.78-3.86) | 7156 (7094) | 2.64 (2.57-2.7) |
| 5 | 33130 (44697) | 4.89 (4.83-4.95) | 3353 (4268) | 3.13 (3.03-3.23) |
| >5 | 33390 (68070) | 7.49 (7.41-7.57) | 2403 (4005) | 4.11 (3.97-4.25) |

**Fig 6. Relationship between LTC count and IRR for unscheduled hospitalisation in UK Biobank and SAIL.** Points indicate IRRs adjusted for age, sex, and socioeconomic status. Whiskers indicate 95% CIs. *N* = total number of participants per LTC count, n events = number of events. CI, confidence interval; IRR, incidence rate ratio; LTC, long-term condition; SAIL, Secure Anonymised Information Linkage.

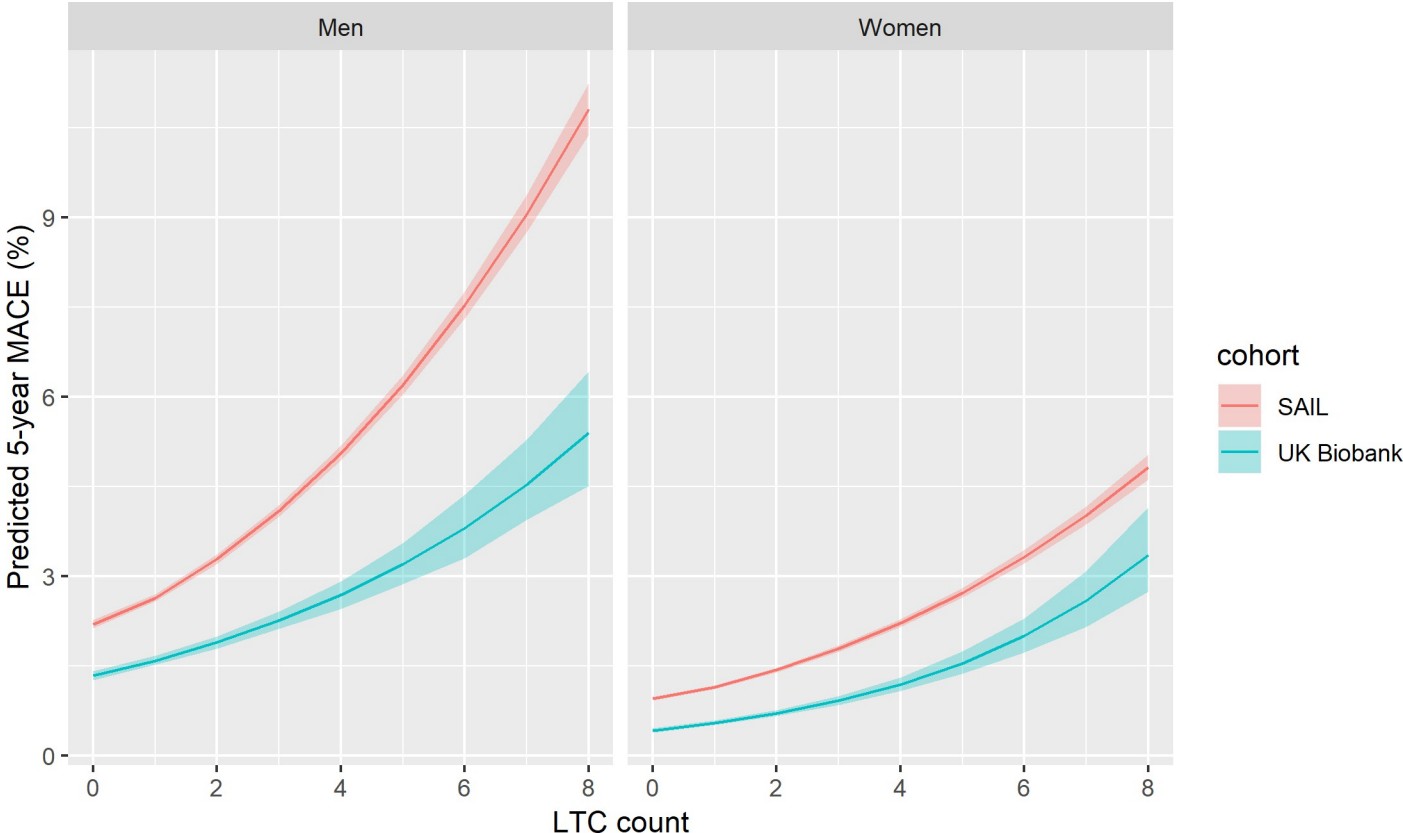

**Fig 7. Relationship between LTC count and absolute risk of MACE for UK Biobank and SAIL.** Lines indicate the modelled 5-year risk of MACE. Shaded area indicates 95% confidence interval. Predicted values are modelled at mean age for UK Biobank (56.5 years) and UK national mean Townsend score for socioeconomic status. LTC, long-term condition; MACE, major adverse cardiovascular event; SAIL, Secure Anonymised Information Linkage.

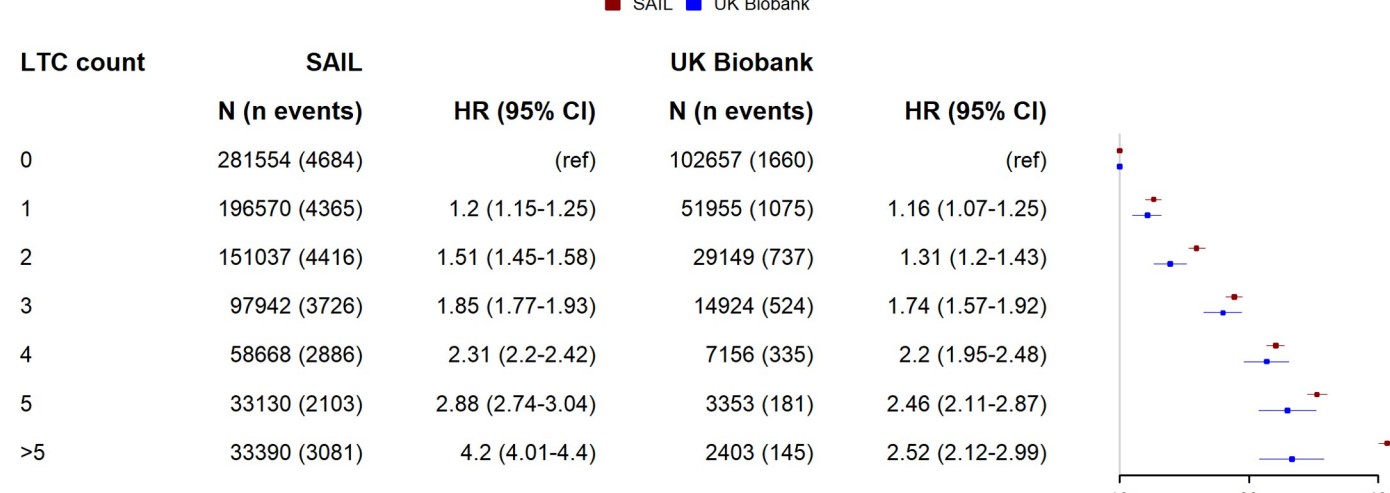

| LTC count | SAIL | | UK Biobank | | |
|---|---|---|---|---|---|
| | N (n events) | HR (95% CI) | N (n events) | HR (95% CI) | |
| 0 | 281554 (4684) | (ref) | 102657 (1660) | (ref) | |
| 1 | 196570 (4365) | 1.2 (1.15-1.25) | 51955 (1075) | 1.16 (1.07-1.25) | |
| 2 | 151037 (4416) | 1.51 (1.45-1.58) | 29149 (737) | 1.31 (1.2-1.43) | |
| 3 | 97942 (3726) | 1.85 (1.77-1.93) | 14924 (524) | 1.74 (1.57-1.92) | |
| 4 | 58668 (2886) | 2.31 (2.2-2.42) | 7156 (335) | 2.2 (1.95-2.48) | |
| 5 | 33130 (2103) | 2.88 (2.74-3.04) | 3353 (181) | 2.46 (2.11-2.87) | |
| >5 | 33390 (3081) | 4.2 (4.01-4.4) | 2403 (145) | 2.52 (2.12-2.99) | |

**Fig 8. Relationship between LTC count and incident rate ratio for unscheduled hospitalisation in UK Biobank and SAIL.** Points indicate HRs adjusted for age, sex, and socioeconomic status. Whiskers indicate 95% CIs. *N* = total number of participants per LTC count, n events = number of events. CI, confidence interval; HR, hazard ratio; LTC, long-term condition; SAIL, Secure Anonymised Information Linkage.

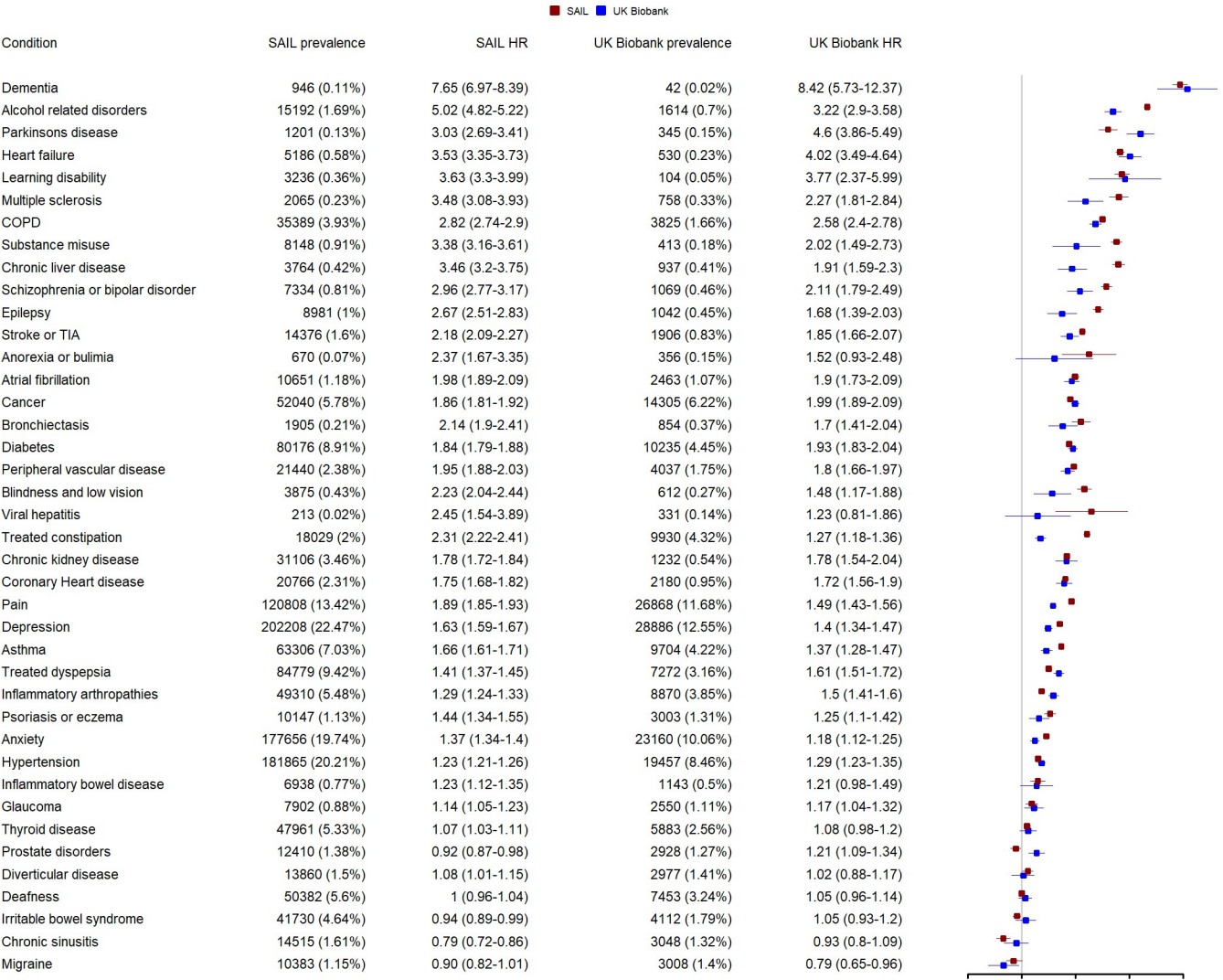

| Condition | SAIL prevalence | SAIL HR | UK Biobank prevalence | UK Biobank HR |
|---|---|---|---|---|
| Dementia | 946 (0.11%) | 7.65 (6.97-8.39) | 42 (0.02%) | 8.42 (5.73-12.37) |
| Alcohol related disorders | 15192 (1.69%) | 5.02 (4.82-5.22) | 1614 (0.7%) | 3.22 (2.9-3.58) |
| Parkinsons disease | 1201 (0.13%) | 3.03 (2.69-3.41) | 345 (0.15%) | 4.6 (3.86-5.49) |
| Heart failure | 5186 (0.58%) | 3.53 (3.35-3.73) | 530 (0.23%) | 4.02 (3.49-4.64) |
| Learning disability | 3236 (0.36%) | 3.63 (3.3-3.99) | 104 (0.05%) | 3.77 (2.37-5.99) |
| Multiple sclerosis | 2065 (0.23%) | 3.48 (3.08-3.93) | 758 (0.33%) | 2.27 (1.81-2.84) |
| COPD | 35389 (3.93%) | 2.82 (2.74-2.9) | 3825 (1.66%) | 2.58 (2.4-2.78) |
| Substance misuse | 8148 (0.91%) | 3.38 (3.16-3.61) | 413 (0.18%) | 2.02 (1.49-2.73) |
| Chronic liver disease | 3764 (0.42%) | 3.46 (3.2-3.75) | 937 (0.41%) | 1.91 (1.59-2.3) |
| Schizophrenia or bipolar disorder | 7334 (0.81%) | 2.96 (2.77-3.17) | 1069 (0.46%) | 2.11 (1.79-2.49) |
| Epilepsy | 8981 (1%) | 2.67 (2.51-2.83) | 1042 (0.45%) | 1.68 (1.39-2.03) |
| Stroke or TIA | 14376 (1.6%) | 2.18 (2.09-2.27) | 1906 (0.83%) | 1.85 (1.66-2.07) |
| Anorexia or bulimia | 670 (0.07%) | 2.37 (1.67-3.35) | 356 (0.15%) | 1.52 (0.93-2.48) |
| Atrial fibrillation | 10651 (1.18%) | 1.98 (1.89-2.09) | 2463 (1.07%) | 1.9 (1.73-2.09) |
| Cancer | 52040 (5.78%) | 1.86 (1.81-1.92) | 14305 (6.22%) | 1.99 (1.89-2.09) |
| Bronchiectasis | 1905 (0.21%) | 2.14 (1.9-2.41) | 854 (0.37%) | 1.7 (1.41-2.04) |
| Diabetes | 80176 (8.91%) | 1.84 (1.79-1.88) | 10235 (4.45%) | 1.93 (1.83-2.04) |
| Peripheral vascular disease | 21440 (2.38%) | 1.95 (1.88-2.03) | 4037 (1.75%) | 1.8 (1.66-1.97) |
| Blindness and low vision | 3875 (0.43%) | 2.23 (2.04-2.44) | 612 (0.27%) | 1.48 (1.17-1.88) |
| Viral hepatitis | 213 (0.02%) | 2.45 (1.54-3.89) | 331 (0.14%) | 1.23 (0.81-1.86) |
| Treated constipation | 18029 (2%) | 2.31 (2.22-2.41) | 9930 (4.32%) | 1.27 (1.18-1.36) |
| Chronic kidney disease | 31106 (3.46%) | 1.78 (1.72-1.84) | 1232 (0.54%) | 1.78 (1.54-2.04) |
| Coronary Heart disease | 20766 (2.31%) | 1.75 (1.68-1.82) | 2180 (0.95%) | 1.72 (1.56-1.9) |
| Pain | 120808 (13.42%) | 1.89 (1.85-1.93) | 26868 (11.68%) | 1.49 (1.43-1.56) |
| Depression | 202208 (22.47%) | 1.63 (1.59-1.67) | 28886 (12.55%) | 1.4 (1.34-1.47) |
| Asthma | 63306 (7.03%) | 1.66 (1.61-1.71) | 9704 (4.22%) | 1.37 (1.28-1.47) |
| Treated dyspepsia | 84779 (9.42%) | 1.41 (1.37-1.45) | 7272 (3.16%) | 1.61 (1.51-1.72) |
| Inflammatory arthropathies | 49310 (5.48%) | 1.29 (1.24-1.33) | 8870 (3.85%) | 1.5 (1.41-1.6) |
| Psoriasis or eczema | 10147 (1.13%) | 1.44 (1.34-1.55) | 3003 (1.31%) | 1.25 (1.1-1.42) |
| Anxiety | 177656 (19.74%) | 1.37 (1.34-1.4) | 23160 (10.06%) | 1.18 (1.12-1.25) |
| Hypertension | 181865 (20.21%) | 1.23 (1.21-1.26) | 19457 (8.46%) | 1.29 (1.23-1.35) |
| Inflammatory bowel disease | 6938 (0.77%) | 1.23 (1.12-1.35) | 1143 (0.5%) | 1.21 (0.98-1.49) |
| Glaucoma | 7902 (0.88%) | 1.14 (1.05-1.23) | 2550 (1.11%) | 1.17 (1.04-1.32) |
| Thyroid disease | 47961 (5.33%) | 1.07 (1.03-1.11) | 5883 (2.56%) | 1.08 (0.98-1.2) |
| Prostate disorders | 12410 (1.38%) | 0.92 (0.87-0.98) | 2928 (1.27%) | 1.21 (1.09-1.34) |
| Diverticular disease | 13860 (1.5%) | 1.08 (1.01-1.15) | 2977 (1.41%) | 1.02 (0.88-1.17) |
| Deafness | 50382 (5.6%) | 1 (0.96-1.04) | 7453 (3.24%) | 1.05 (0.96-1.14) |
| Irritable bowel syndrome | 41730 (4.64%) | 0.94 (0.89-0.99) | 4112 (1.79%) | 1.05 (0.93-1.2) |
| Chronic sinusitis | 14515 (1.61%) | 0.79 (0.72-0.86) | 3048 (1.32%) | 0.93 (0.8-1.09) |
| Migraine | 10383 (1.15%) | 0.90 (0.82-1.01) | 3008 (1.4%) | 0.79 (0.65-0.96) |

**Fig 9. Relationship between each LTC and all-cause mortality in UK Biobank and SAIL.** Points indicate HRs adjusted for age, sex, and socioeconomic status. Whiskers indicate 95% CIs. Conditions are ordered by the mean HR (of UK Biobank and SAIL) for each condition. CI, confidence interval; COPD, chronic obstructive pulmonary disease; HR, hazard ratio; LTC, long-term condition; SAIL, Secure Anonymised Information Linkage; TIA, transient ischaemic attack.

Biobank. However, a few conditions (e.g., multiple sclerosis, anorexia or bulimia, and Parkinson's disease) were slightly more common in UK Biobank than in SAIL.

For many common conditions, the hazard ratios for all-cause mortality were similar between UK Biobank and SAIL (e.g., chronic kidney disease, coronary heart disease). Other conditions had notably higher hazard ratios in SAIL than in UK Biobank (e.g., alcohol-related disorders, illicit drug use, and chronic liver disease).

## Association between combinations of LTCs and clinical outcomes

The relationship between specific combinations of LTCs and each outcome (mortality, hospitalisation, and MACE) are shown in the Supporting information (S11–S34 Figs). Associations between LTC combinations and outcomes were similar between the 2 datasets for some LTC types (most notably multiple cardiovascular conditions, or cardiovascular and respiratory

conditions). Risk of outcomes was underestimated in UK Biobank for other combinations, particularly those including depression, anxiety, or pain (S11–S34 Figs).

## Sensitivity analyses

In a sensitivity analysis of the relationship between LTC count and each outcome (all-cause mortality, hospitalisations, and MACE), we restricted UK Biobank participants to those attending assessment centres in Wales. The adjusted event rates were slightly lower in the Welsh UK Biobank subset than in the whole UK Biobank cohort (S35–S37 Figs). Separately, we restricted the SAIL analyses to Cardiff and Swansea (the location of UK Biobank assessment centres), finding similar rates to the main analysis for mortality and MACE. For hospitalisation, the difference between SAIL participants from Cardiff and Swansea and UK Biobank participants from Wales was less than the difference between each of the full cohorts (S36 Fig), although rates in SAIL remained higher than UK Biobank. This suggests that differences between SAIL and UK Biobank are unlikely to be driven by differences between Wales and the rest of the UK.

## Discussion

Multimorbidity in UK Biobank, as identified through linked primary care Read code data, is both less common and associated with a lower absolute risk of all-cause mortality than in a similarly aged representative sample from Wales (SAIL databank), whether one assesses multimorbidity via a simple count or uses a weighted scale that takes type of LTC into account. Differences in socioeconomic deprivation account for some, but not all, of the observed difference in the distribution of multimorbidity. The difference in absolute mortality risk persisted after adjustment for socioeconomic deprivation. Despite this, the relative association between multimorbidity and mortality and hospitalisation seen in UK Biobank were comparable to SAIL at lower LTC counts (below 4). Moreover, although generally less prevalent in UK Biobank, many common LTCs such as hypertension, chronic kidney disease, chronic obstructive pulmonary disease, and coronary heart disease had a similar relative risk of mortality between the 2 datasets. These findings suggest that, given its large sample size and extensive baseline assessment, UK Biobank is a valuable resource for exploring risks associated with chronic illness or multimorbidity with lower LTC counts. However, UK Biobank is likely to underestimate the impact of multimorbidity in people with greater numbers of LTCs or with specific LTCs such as mental health conditions. This difference was most obvious among participants with LTC counts or 4 or more.

There are a number of possible explanations for the lower risk of mortality and hospitalisation associated with a given LTC count in UK Biobank. First, the specific LTCs present in UK Biobank participants with multimorbidity are likely to differ, particularly as we found that the differences in prevalence are not evenly distributed across all LTCs (e.g., dementia, schizophrenia, and illicit drug use are particularly underrepresented). This is unsurprising as many of these conditions are likely to impact the likelihood of volunteering for recruitment to this type of study, and show strong relationships with mortality. However, the difference in mortality and hospitalisation was similar when using the Cambridge multimorbidity score, which weights each LTC by its respective risk of each outcome, suggesting that type of LTC alone cannot explain the observed difference.

Second, LTCs in UK Biobank may be less severe or advanced than in the population in general. This could occur if severe disease resulted in increased functional limitation which in turn reduced the likelihood people would volunteer to attend for UK Biobank assessment. This hypothesis is supported by our findings for conditions such as chronic liver disease, in

which the prevalence was similar in UK Biobank and SAIL, but the hazard ratio for mortality was 2-fold higher in SAIL compared to UK Biobank. Unfortunately, it is usually not possible to estimate severity of LTCs using routine data alone.

Finally, additional factors are likely to influence the risk of adverse outcomes such as behavioural and environmental risk factors. Also, area-based measures of socioeconomic deprivation may not pick up all relevant elements. Within a given area (within which individuals will be assigned the same value for socioeconomic status), there is likely to be variation in factors such as income, education, disability, and behaviours such as smoking and alcohol consumption. These factors may influence the likelihood of volunteering for studies such as UK Biobank, but may not be adequately reflected in an area-based measure of socioeconomic status. UK Biobank participants are recognised to have lower levels of various lifestyle risk factors, such as smoking, than the UK population, and this is likely to mitigate the impact of multimorbidity [18]. Indeed, previous UK Biobank analyses have demonstrated that lifestyle factors influence life expectancy in people with multimorbidity [11]. Also, unhealthy lifestyle factors disproportionately impact mortality risk in the context of higher socioeconomic deprivation [29]. Selection bias of UK Biobank on these features is therefore likely to influence the relationship between multimorbidity and adverse health outcomes.

Compared to individuals without LTCs, the association between LTC count and outcome was more similar across SAIL and UK Biobank at lower counts (e.g., 1, 2, 3 conditions) for both death and hospitalisation. One possible explanation for this is differential selection pressures. As the number of LTCs increase, individuals able to undergo the UK Biobank visits and procedures are likely to be increasingly atypical of individuals in the community with that number of conditions (e.g., they may have less severe disease or less functional impairment). Similarly, for certain conditions (such as mental health conditions or addiction, where the difference in prevalence is large), people volunteering for UK Biobank may be less typical of people with these conditions in the community. It is also possible that, for some people, living with a condition may be a motivation for participation in UK Biobank. Furthermore, some LTCs have been noted to be more common in more affluent populations [30]. We speculate that this may be one possible explanation for the higher prevalence of some conditions (e.g., Parkinson's disease) in UK Biobank compared to SAIL.

Previous studies have compared associations in UK Biobank to national surveys with higher response rates (e.g., Health Survey for England and Health Survey for Scotland) [18,21]. To our knowledge, this is the first study to directly compare estimates from UK Biobank with those from routinely collected data from a representative sample, and also the first to do so for multimorbidity. Batty and colleagues reported that associations between a wide range of established risk factors and mortality were similar between UK Biobank and more representative surveys, suggesting that associations were largely generalizable and reassured that UK Biobank was a useful resource for studying aetiological mechanisms [21]. However, like UK Biobank, surveys may involve visits and procedures, and so biases based on health and functional status are likely to operate in both settings. Furthermore, for some risk factors, differences were observed between estimates from UK Biobank and more representative surveys.

While some of our findings, such as the relationship between specific LTCs and mortality, support this previous study [21], other findings highlight important caveats relating to multimorbidity research. First, some underrepresented conditions may not give reliable estimates. Second, at higher LTC counts (where UK Biobank data is sparce), estimates become less reliable. Finally, assessing absolute event rates is particularly likely to yield underestimates.

A previous study assessing the impact of cardiometabolic multimorbidity analysed UK Biobank alongside pooled data from 91 cohorts in the Emerging Risk Factors collaboration [12]. This study found that diabetes, stroke, and coronary heart disease (alone or in combination)

were associated with similar risks across these datasets. We found similar results for these common conditions, both alone and in combination. However, we found greater differences with other conditions, which are underrepresented, such as mental health conditions, addiction, and pain.

Previous studies of multimorbidity using UK Biobank have acknowledged the lack of representativeness and cautioned that estimates of association are likely to be conservative [4,9]. Our findings support this hypothesis, suggesting that both relative and absolute risks of multimorbidity estimated using UK Biobank are likely to be lower than in the population in general. Moreover, where multimorbidity counts are included as a potential confounder, rather than as the exposure of interest, the direction of effect may be less predictable.

Observational studies are crucial to advancing our understanding of multimorbidity, particularly as people with multimorbidity are often excluded from clinical trials [25,31]. Furthermore, UK Biobank contains a wealth of phenotypic and lifestyle data that is not discoverable from routine data alone. In addition, follow-up questionnaires facilitate analysis of the relationships between multimorbidity and diverse factors not recorded in routine data [32]. Therefore, studies such as UK Biobank offer valuable opportunities to understand the causes, phenotypic manifestation, and wide-ranging implications of multimorbidity. However, our findings reinforce the need for caution in interpreting findings, particularly as some estimates are likely to be conservative. Furthermore, people with multimorbidity with high LTC counts are underrepresented in UK Biobank: 1.1% of UK Biobank participants had LTC counts greater than 5, compared to 3.8% of people in SAIL. While these people are a small proportion of the overall population, many clinical and public health challenges are concentrated and intensified in such individuals.

Potential for selection bias is a problem for cohort studies in general, not just UK Biobank. This study emphasises the importance of assessing population demographics compared to the general population and being mindful of the potential for estimates to be biased by a lack of representativeness. Our findings suggest that UK Biobank may be biased (and likely conservative) in making inferences for the population of people with multimorbidity involving high LTC counts, many of whom live in areas of high socioeconomic deprivation. However, the impact of selection bias is unpredictable, and this pattern may not necessarily be the same in other cohorts susceptible to selection bias. In the absence of a direct comparison with a representative sample, investigators should be cautious in their interpretation of findings derived from highly selected cohorts. This suggests a need for future research cohorts to intensify efforts to target recruitment to ensure people with multimorbidity or living in areas of high socioeconomic deprivation are included. It will also be worth considering how routine healthcare data can be linked with other information resources or enhanced by collection of additional data to enable more mechanistic research for those with complex multimorbidity and those living in areas of high socioeconomic deprivation [33].

Strengths of this analysis include the use of Read coded primary care data to identify multimorbidity in both data sources. This allowed identical definitions to be applied to both UK Biobank and SAIL. The use of linked hospitalisation and mortality data allowed reliable estimation of outcomes. Therefore, our analyses are likely to give an accurate representation of the differences in multimorbidity and association with outcomes between UK Biobank and an unselected community population from within the UK. However, some conditions, such as chronic pain, may be underidentified by using Read coded data compared with self-report. Also, SAIL includes data from Wales only, whereas UK Biobank comprises England, Scotland, and Wales, although this is unlikely to have a substantial impact on our findings. There may also be some overlap between UK Biobank participants in Wales and people included in SAIL databank; however, we were unable to identify UK Biobank participants in SAIL. As discussed

above, we lacked data on the severity of the included LTCs (in both datasets), as well as data on ethnicity and broader lifestyle factors from SAIL. It is therefore not possible to disentangle the relative impact of LTC type, severity, and wider factors on the differences in mortality and hospitalisation risk. It is likely that the differences in association seen are a combination of these factors. Finally, we were only able to analyse a subset of the UK Biobank cohort as, to date, primary care data linkage is only available for this subset. However, these participants are similar to the wider cohort in terms of age, sex, socioeconomic status, and prevalence of multimorbidity (S1 Table).

In conclusion, UK Biobank accurately represents the increased relative risks associated with multimorbidity involving LTC counts of 3 or less and therefore can be a useful resource for multimorbidity research. However, its lack of representativeness limits inferences about the minority of people with higher levels of multimorbidity, most noticeably those with LTC counts of 4 or more. The rich lifestyle, environmental, phenotypic, and genetic data available in UK Biobank offers unique opportunities to understand factors associated with development of multimorbidity as well as relationships between multimorbidity and various associated phenomena. Nonetheless, researchers should understand that where LTCs are the exposure, UK Biobank is likely to provide results that are more conservative than one would find in a general population cohort. Ideally, future research should combine insights from both representative routine data and information rich research cohorts such as UK Biobank.

## Declarations

### Ethics committee approval

This study had ethical approval as part of UK Biobank project 14151 (NHS National Research Ethics Service 16/NW/0274). SAIL analyses were approved by SAIL Information Governance Review Panel (Project 0830).

### Dissemination

Published findings will be returned to UK Biobank, from whom findings are made available to UK Biobank participants via the UK Biobank website.

## Supporting information

**S1 Protocol. Analysis plan.**
(DOCX)

**S1 Checklist. STROBE reporting checklist.**
(DOCX)

**S1 File. Read codes for included LTCs.** LTC, long-term condition.
(CSV)

**S2 File. Read codes for drugs used in LTC definitions.** LTC, long-term condition.
(CSV)

**S1 Table. Comparison of UK Biobank participants with linked GP data versus those without GP data available.** GP, General Practice.
(DOCX)

**S2 Table. Observed and expected LTC counts in UK Biobank.** Includes confidence intervals pertaining to Fig 2. LTC, long-term condition.
(DOCX)

**S3 Table. Observed event rates in UK Biobank and SAIL.** Observed rates of all-cause mortality, unscheduled hospitalisation, and MACE in UK Biobank and SAIL, by number of baseline LTCs. LTC, long-term condition; MACE, major adverse cardiovascular event; SAIL, Secure Anonymised Information Linkage.
(DOCX)

**S4 Table. Unadjusted hazard ratios.** Includes unadjusted hazard ratios for those presented in the main analysis.
(DOCX)

**S5 Table. Cambridge multimorbidity score: Comparison of UK Biobank and SAIL.** Descriptive statistics of Cambridge multimorbidity score in UK Biobank and SAIL. SAIL, Secure Anonymised Information Linkage.
(DOCX)

**S1 Fig. Model diagnostic for SAIL LTC model.** Observed and expected LTC counts in SAIL, stratified by age, sex, and socioeconomic status. LTC, long-term condition; SAIL, Secure Anonymised Information Linkage.
(PDF)

**S2 Fig. Comparison of model fit: All-cause mortality in UK Biobank and SAIL.** Line indicates the modelled values for each cohort (stratified by age), shaded area indicates 95% CIs, points indicate the observed proportion of deaths within each stratum of age and LTC count. Size of the point indicates the number of events per strata. CI, confidence interval; LTC, long-term condition; SAIL, Secure Anonymised Information Linkage.
(PDF)

**S3 Fig. Comparison of model fit: MACE in UK Biobank and SAIL.** Line indicates the modelled values for each cohort (stratified by age), shaded area indicates 95% CIs, points indicate the observed proportion of deaths within each stratum of age and LTC count. Size of the point indicates the number of events per strata. CI, confidence interval; LTC, long-term condition; MACE, major adverse cardiovascular event; SAIL, Secure Anonymised Information Linkage.
(PDF)

**S4 Fig. Model fit, negative binomial model for unscheduled hospitalisations.** Observed and expected number of unscheduled hospitalisations. SAIL, Secure Anonymised Information Linkage.
(PDF)

**S5 Fig. Model fit, negative binomial model for unscheduled hospitalisations, stratified by age, sex, and socioeconomic status.** Observed and expected number of unscheduled hospitalisations. LTC, long-term condition; SAIL, Secure Anonymised Information Linkage.
(PDF)

**S6 Fig. Association of multimorbidty and all-cause mortality by age, sex, and socioeconomic status.** Line indicates the modelled values for each cohort; shaded area indicates 95% CIs. Plots are stratified by age, sex, and socioeconomic status quintiles. CI, confidence interval; SAIL, Secure Anonymised Information Linkage.
(PDF)

**S7 Fig. Cambridge score and all-cause mortality.** Line indicates the modelled values for each cohort; shaded area indicates 95% CIs. CI, confidence interval; SAIL, Secure Anonymised

Information Linkage.
(PDF)

**S8 Fig. Association of multimorbidty and unscheduled hospitalisation by age, sex, and socioeconomic status.** Line indicates the modelled values for each cohort; shaded area indicates 95% CIs. Plots are stratified by age, sex, and socioeconomic status quintiles. CI, confidence interval; SAIL, Secure Anonymised Information Linkage.
(PDF)

**S9 Fig. Cambridge score and unscheduled hospitalisation.** Line indicates the modelled values for each cohort; shaded area indicates 95% CIs. CI, confidence interval; SAIL, Secure Anonymised Information Linkage.
(PDF)

**S10 Fig. Association of multimorbidty and MACE by age, sex, and socioeconomic status.** Line indicates the modelled values for each cohort; shaded area indicates 95% CIs. Plots are stratified by age, sex, and socioeconomic status quintiles. CI, confidence interval; MACE, major adverse cardiovascular event; SAIL, Secure Anonymised Information Linkage.
(PDF)

**S11 Fig. HRs for all-cause mortality: Coronary heart disease, diabetes, and stroke.** CI, confidence interval; HR, hazard ratio; SAIL, Secure Anonymised Information Linkage; TIA, transient ischaemic attack; UKB, UK Biobank.
(PNG)

**S12 Fig. HRs for all-cause mortality: Hypertension, diabetes, and pain.** CI, confidence interval; HR, hazard ratio; SAIL, Secure Anonymised Information Linkage; UKB, UK Biobank.
(PNG)

**S13 Fig. HRs for all-cause mortality: Irritable bowel syndrome, deafness, and pain.** CI, confidence interval; HR, hazard ratio; SAIL, Secure Anonymised Information Linkage; UKB, UK Biobank.
(PNG)

**S14 Fig. HRs for all-cause mortality: Depression, pain, and anxiety.** CI, confidence interval; HR, hazard ratio; SAIL, Secure Anonymised Information Linkage; UKB, UK Biobank.
(PNG)

**S15 Fig. HRs for all-cause mortality: Asthma, pain, and COPD.** CI, confidence interval; COPD, chronic obstructive pulmonary disease; HR, hazard ratio; SAIL, Secure Anonymised Information Linkage; UKB, UK Biobank.
(PNG)

**S16 Fig. HRs for all-cause mortality: Alcohol-related disorders, illicit drug use, and pain.** CI, confidence interval; HR, hazard ratio; SAIL, Secure Anonymised Information Linkage; UKB, UK Biobank.
(PNG)

**S17 Fig. HRs for all-cause mortality: Coronary heart disease, diabetes, and atrial fibrillation.** CI, confidence interval; HR, hazard ratio; SAIL, Secure Anonymised Information Linkage; UKB, UK Biobank.
(PNG)

**S18 Fig. HRs for all-cause mortality: Pain, coronary heart disease, and depression.** CI, confidence interval; HR, hazard ratio; SAIL, Secure Anonymised Information Linkage; UKB, UK Biobank.
(PNG)

**S19 Fig. IRRs for unscheduled hospitalisation: Coronary heart disease, diabetes, and stroke.** CI, confidence interval; IRR, incidence rate ratio; SAIL, Secure Anonymised Information Linkage; TIA, transient ischaemic attack; UKB, UK Biobank.
(PNG)

**S20 Fig. IRRs for unscheduled hospitalisation: Hypertension, diabetes, and pain.** CI, confidence interval; IRR, incidence rate ratio; SAIL, Secure Anonymised Information Linkage; UKB, UK Biobank.
(PNG)

**S21 Fig. IRRs for unscheduled hospitalisation: Irritable bowel syndrome, deafness, and pain.** CI, confidence interval; IRR, incidence rate ratio; SAIL, Secure Anonymised Information Linkage; UKB, UK Biobank.
(PNG)

**S22 Fig. IRRs for unscheduled hospitalisation: Depression, pain, and anxiety.** CI, confidence interval; IRR, incidence rate ratio; SAIL, Secure Anonymised Information Linkage; UKB, UK Biobank.
(PNG)

**S23 Fig. IRRs for unscheduled hospitalisation: Asthma, pain, and COPD.** CI, confidence interval; COPD, chronic obstructive pulmonary disease; IRR, incidence rate ratio; SAIL, Secure Anonymised Information Linkage; UKB, UK Biobank.
(PNG)

**S24 Fig. IRRs for unscheduled hospitalisation: Alcohol-related disorders, illicit drug use, and pain.** CI, confidence interval; IRR, incidence rate ratio; SAIL, Secure Anonymised Information Linkage; UKB, UK Biobank.
(PNG)

**S25 Fig. IRRs for unscheduled hospitalisation: Coronary heart disease, diabetes, and atrial fibrillation.** CI, confidence interval; IRR, incidence rate ratio; SAIL, Secure Anonymised Information Linkage; UKB, UK Biobank.
(PNG)

**S26 Fig. IRRs for unscheduled hospitalisation: Pain, coronary heart disease, and depression.** CI, confidence interval; IRR, incidence rate ratio; SAIL, Secure Anonymised Information Linkage; UKB, UK Biobank.
(PNG)

**S27 Fig. HRs for MACE: Coronary heart disease, diabetes, and stroke.** CI, confidence interval; HR, hazard ratio; MACE, major adverse cardiovascular event; SAIL, Secure Anonymised Information Linkage; TIA, transient ischaemic attack; UKB, UK Biobank.
(PNG)

**S28 Fig. HRs for MACE: Hypertension, diabetes, and pain.** CI, confidence interval; HR, hazard ratio; MACE, major adverse cardiovascular event; SAIL, Secure Anonymised Information Linkage; UKB, UK Biobank.
(PNG)

**S29 Fig. HRs for MACE: Irritable bowel syndrome, deafness, and pain.** CI, confidence interval; HR, hazard ratio; MACE, major adverse cardiovascular event; SAIL, Secure Anonymised Information Linkage; UKB, UK Biobank.
(PNG)

**S30 Fig. HRs for MACE: Depression, pain, and anxiety.** CI, confidence interval; HR, hazard ratio; MACE, major adverse cardiovascular event; SAIL, Secure Anonymised Information Linkage; UKB, UK Biobank.
(PNG)

**S31 Fig. HRs for MACE: Asthma, pain, and COPD.** CI, confidence interval; COPD, chronic obstructive pulmonary disease; HR, hazard ratio; MACE, major adverse cardiovascular event; SAIL, Secure Anonymised Information Linkage; UKB, UK Biobank.
(PNG)

**S32 Fig. HRs for MACE: Alcohol-related disorders, illicit drug use, and pain.** CI, confidence interval; HR, hazard ratio; MACE, major adverse cardiovascular event; SAIL, Secure Anonymised Information Linkage; UKB, UK Biobank.
(PNG)

**S33 Fig. HRs for MACE: Coronary heart disease, diabetes, and atrial fibrillation.** CI, confidence interval; HR, hazard ratio; MACE, major adverse cardiovascular event; SAIL, Secure Anonymised Information Linkage; UKB, UK Biobank.
(PNG)

**S34 Fig. HRs for MACE: Pain, coronary heart disease, and depression.** CI, confidence interval; HR, hazard ratio; MACE, major adverse cardiovascular event; SAIL, Secure Anonymised Information Linkage; UKB, UK Biobank.
(PNG)

**S35 Fig. Sensitivity analysis: All-cause mortality.** Comparison of full cohort and area-specific estimates. LTC, long-term condition; SAIL, Secure Anonymised Information Linkage.
(PDF)

**S36 Fig. Sensitivity analysis: Unscheduled hospitalisation.** Comparison of full cohort and area-specific estimates. LTC, long-term condition; SAIL, Secure Anonymised Information Linkage.
(PDF)

**S37 Fig. Sensitivity analysis: MACE.** Comparison of full cohort and area-specific estimates. LTC, long-term condition; MACE, major adverse cardiovascular event; SAIL, Secure Anonymised Information Linkage.
(PDF)

## Author Contributions

**Conceptualization:** Peter Hanlon, Bhautesh D. Jani, Jim Lewsey, Frances S. Mair.

**Data curation:** Peter Hanlon.

**Formal analysis:** Peter Hanlon, David A. McAllister.

**Funding acquisition:** Peter Hanlon.

**Methodology:** Jim Lewsey, David A. McAllister.

**Project administration:** Peter Hanlon.

**Supervision:** Jim Lewsey, David A. McAllister, Frances S. Mair.

**Visualization:** Peter Hanlon.

**Writing – original draft:** Peter Hanlon.

**Writing – review & editing:** Peter Hanlon, Bhautesh D. Jani, Barbara Nicholl, Jim Lewsey, David A. McAllister, Frances S. Mair.

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
