## [Decision Letter · Decision Letter 0]

13 Sep 2021

Dear Dr. Hanlon,

Thank you very much for submitting your manuscript "Associations between multimorbidity and adverse health outcomes: comparison of cohorts from UK Biobank and a representative community sample" (PMEDICINE-D-21-02504) for consideration at PLOS Medicine. 

Your paper was evaluated by three independent reviewers, including a statistical reviewer, and was discussed among all the editors here and with an academic editor with relevant expertise. The reviews are appended at the bottom of this email and any accompanying reviewer attachments can be seen via the link below:

[LINK]

In light of these reviews, I am afraid that we will not be able to accept the manuscript for publication in the journal in its current form, but we would like to consider a revised version that addresses the reviewers' and editors' comments. Obviously we cannot make any decision about publication until we have seen the revised manuscript and your response, and we plan to seek re-review by one or more of the reviewers. 

We expect to receive your revised manuscript by Oct 04 2021 11:59PM. Please email us (plosmedicine@plos.org) if you have any questions or concerns.

We look forward to receiving your revised manuscript. 

Sincerely,

Louise Gaynor-Brook, MBBS PhD

Associate Editor 

PLOS Medicine

plosmedicine.org

General comments:

Throughout the paper, please adapt reference call-outs to the following style: "... adverse health outcomes [4,5]." (noting the absence of spaces within the square brackets).

Title: Please revise your title according to PLOS Medicine's style, placing the study design in the subtitle (ie, after a colon). We suggest “Associations between multimorbidity and adverse health outcomes in UK Biobank and the SAIL Databank: A comparison of longitudinal cohort studies” or similar

Abstract:

Abstract Methods and Findings:

Please include the study design and years during which data was collected for the UKBB / SAIL cohorts

Please provide brief demographic details of the study population (e.g. sex, age, ethnicity, etc) in each of the databanks studied.

Please quantify the results presented in your abstract (with 95% CIs), ensuring that all numbers presented in the abstract are identical to numbers presented in the main manuscript text.

In the last sentence of the Abstract Methods and Findings section, please describe 2-3 of the main limitations of the study's methodology.

Abstract Conclusions:

Please begin your Abstract Conclusions with "In this study, we observed ..." or similar, to summarize the main findings from your study, without overstating your conclusions. Please emphasize what is new and address the implications of your study, being careful to avoid assertions of primacy. Please be more specific as to which outcomes you are referring to.

Author Summary:

In the final bullet point of ‘What Do These Findings Mean?’, please describe the main limitations of the study in non-technical language.

Introduction:

Please explain the potential importance of your study and indicate whether your study is novel, being careful to temper assertions of primacy. 

Methods:

Did your study have a prospective protocol or analysis plan? Please state this (either way) early in the Methods section. If a prospective analysis plan (from your funding proposal, IRB or other ethics committee submission, study protocol, or other planning document written before analyzing the data) was used in designing the study, please include the relevant prospectively written document with your revised manuscript as a Supporting Information file to be published alongside your study, and cite it in the Methods section. A legend for this file should be included at the end of your manuscript. If no such document exists, please make sure that the Methods section transparently describes when analyses were planned, and if/when reported analyses differed from those that were planned. Changes in the analysis-- including those made in response to peer review comments-- should be identified as such in the Methods section of the paper, with rationale. If a reported analysis was performed based on an interesting but unanticipated pattern in the data, please be clear that the analysis was data-driven.

Please ensure that the study is reported according to the STROBE guideline, and include the completed STROBE checklist as Supporting Information. Please add the following statement, or similar, to the Methods: "This study is reported as per the Strengthening the Reporting of Observational Studies in Epidemiology (STROBE) guideline (S1 Checklist)." The STROBE guideline can be found here: http://www.equator-network.org/reporting-guidelines/strobe/ When completing the checklist, please use section and paragraph numbers, rather than page numbers which will likely no longer correspond to the appropriate sections after copy-editing.

Please provide an ethics statement in your Methods section. 

Results: 

Please provide a table showing the baseline characteristics of the study population (Table 1). 

Please note that figures and tables in the Supplementary Information cannot be grouped together in one document (https://journals.plos.org/plosmedicine/s/supporting-information). Please refer to a specific figure or table e.g. Figure S1, Table S1, etc. throughout the main text. 

Please report percentages to at least one decimal place.

Please present numerators and denominators for percentages, at least in the Tables [not necessarily each time they're mentioned].

Line 224 - please clarify what is represented by the numbers in brackets

Where adjusted analyses are presented, please also provide the unadjusted analyses (this may be in a Table) and please indicate which factors are adjusted for. 

Where possible, please provide the actual numbers of events for the outcomes (this may be in a Table), not just HRs.

Discussion:

Please present and organize the Discussion as follows: a short, clear summary of the article's findings; what the study adds to existing research and where and why the results may differ from previous research; strengths and limitations of the study; implications and next steps for research, clinical practice, and/or public policy; one-paragraph conclusion.

Please remove all subheadings within your Discussion e.g. Strengths and Limitations, Conclusion

Figures:

Please provide titles and legends for all figures (including those in Supporting Information files).

Please define all abbreviations used in the respective figure legend. 

Fig 3a, 4a - please indicate what is represented by the coloured areas and central lines in your graphs; 95% CI?

Fig 3b - please clarify whether adjusted HRs are shown. Please indicate in the figure caption the meaning of the boxes and whiskers shown for UKBB ; why are these not provided for SAIL data?

References:

Please ensure that journal name abbreviations match those found in the National Center for Biotechnology Information (NCBI) databases, and are appropriately formatted and capitalised.

Please also see https://journals.plos.org/plosmedicine/s/submission-guidelines#loc-references for further details on reference formatting. 

Supplementary files: 

Please provide titles and legends for each individual table and figure in the Supporting Information.

Please note that figures and tables in the Supporting Information cannot be grouped together in one document. Please see https://journals.plos.org/plosmedicine/s/supporting-information for our supporting information guidelines. 

Comments from the reviewers:

Reviewer #1: "Associations between multimorbidity and adverse health outcomes: comparison of cohorts from UK Biobank and a representative community sample" compares two datasets: a UK Biobank cohort (n=211,597), and a Wales-based SAIL databank (n=852,055) covering some 70% of the Welsh population, for multimorbidity on n=40 long-term conditions (LTC). Various adjusted associations were examined with both unweighted and weighted (Cambridge) scores, and the datasets were found to be similar for associations between multimorbidity and various outcomes when LTC<=3, i.e. UK Biobank appears broadly representative for such cases. However, for LTC>3, UK Biobank estimates were found to likely be conservative. Some plausible explanations for the divergence in observed associations for LTC>3 were discussed.

The analysis appears generally well-executed, and should help to justify the validity of UK Biobank cohort data for research involving populations with relatively low multimorbidities. A number of comments follow:

1. It might be briefly commented as to whether the LTC definitions, and LTC classification by medical personnel, is likely to be applied very similarly for both UK Biobank and SAIL, i.e. are there any significant variances in medical practice between the component countries of the U.K.?

2. It might also be commented as to whether there exists possible overlaps between UK Biobank and SAIL data, for Welsh participants, i.e. might some Welsh participants be considered as part of both datasets?

3. Related to the above, it might be interesting to consider country-level sensitivity analyses if convenient/possible for UK Biobank - in particular UK Biobank (Wales) - vs. SAIL.

4. In Line 147, it is stated that "Age was taken as age at recruitment for UK Biobank, and calculated for 1st January 2011 for SAIL participants". It might be clarified as to whether the data involved must correspond to age at recruitment (for UK Biobank participants) or to 2011 (for SAIL participants); in particular, is it possible that relatively out-of-date data is involved in the analysis (e.g. SAIL participant had last check-up in say 2008, but those results are reported with his age in 2011)?

5. In Line 172, it is stated that "We used fractional polynomials to model non-linear relationships between age and Townsend score and LTC counts". The specific methodology might be described in supplementary material and/or cited as appropriate.

6. In Line 184, it is stated that "Weibull models including fractional polynomials for age and Townsend scores, as well as interaction terms between LTC count and age, and LTC count and sex, were found to fit the data best". It might be clarified as to what this "best" is compared against.

7. In general, for the Statistical Analysis section, it might be considered to state the (Supplementary) Figure/Table corresponding to each analysis, and label the Supplementary Figures/Tables accordingly.

8. For Figure 2, it might be illustrative to also display raw unadjusted counts for SAIL, if possible.

Reviewer #2: 

The study covers an important topic in multimorbidity research, with most likely implications for other cohort studies (also beyond the UK). The methods and results are well presented with systematic comparison with various methods and outcomes. I enjoyed reading the paper.

Major comments:

1) how did the authors handle differences in follow-up time, or completeness of follow-up across datasets? I can image that differences in attrition or (predicted) time-horizons may contribute to differences in absolute risks of adverse health outcomes over time.

2) The authors restricted their analyses to a sample of 40 to 70 years old adults. Why was this done? (40-70 was baseline recruiting age for UK Biobank, but I suppose accrued follow-up now also allows for broader analyses on an age-scale)

3) I commend the authors with their great description of their sophisticated statistical analyses, that is provided in great detail.

4) It may perhaps be useful to readers to know why a cut-off of 3 for LTCs was chosen to compare findings across studies?. Especially since absolute risks for mortality and hospitalization between the two datasets are systematically higher in SAIL compared to UK biobank, regardless of multimorbidity count. Addition of an analysis plan that describes these analyses would benefit the paper, as well as adhering to STROBE guidelines (including for example the checklist)

5) what were reasons to predict the risk of MACE instead of the actual observation risk? Using the cumulative incidence function, it is perhaps more useful to simply show the observed (absolute) rates. Introducing the prediction term, introduces potential confounding of model misspecification or lack of generalizaiblity of the underlying predictive model. 

6) in the methods section, SAIL is introduced as a representative sample of Wales. 70% of the population is covered. Can the authors provide some information on the missing 30% of people? Are they by any chance expected to be any different from the ones that are included?

1) When presenting the prevalence of individual LTCs in UK Biobank and SAIL, the authors note that most LTCs were more prevalent in SAIL, but some LTCs (eg. Parkinsons' disease) were more common in UK Biobank. I'd be helpful to discuss possible explanations for this.

7) The authors hypothesise that LTCs in UK Biobank may be less severe or advanced which would explain the difference in risks of mortality / hospitalisation, but did not offer any explanation as to why LTCs in UK Biobank might be less severe. Since UK Biobank offers information about participants beyond the routine data, it might be possible to explore this further and try to support this hypothesis by assessing the severity of LTCs using those extra information.

8) In the supplementary results, the authors elegantly stratified for depreviation scores. It however seems that regardless of these scores, risks for adverse health outcomes are systematically higher for SAIL compared to UK Biobank participants. Could the authors speculate what drove these higher risks, apparently something beyond social-economic status, age or sex.

Minor comments:

1) In the introduction section, the authors introduce the term collider bias. Not all readers may be familiar with this term, could the authors briefly explain this in the introduction section. 

2) Figure 1 depicts the distribution of LTCs. The authors subsequently report that the mean is higher in SAIL than in UK-biobank. However, this distribution is highly skewed, and I would either suggest to transform it to a normal distribution or report median or range instead of mean.

3) Figure 2. The triangles and circles are relatively small compared to the bars, perhaps enlarge to increase readability. 

4) can the authors clarify to readers what this score exactly means? "Cambridge mortality score of 12 [mean value in UK Biobank], predicted 5-year mortality was 2.2% (2.1 to 2.3) in UK Biobank and 4.0% (4.0-4.1) in SAIL, see supplementary appendix)." For example, what kind of combinations does one then have?

5) It would be very helpful to discuss the generalizability of the findings to other cohort studies with risk of selection bias, and its implications for multimorbidity studies in general.

6) The authors might consider adding confidence intervals to the modelled counts in Figure 2.

7) the results section contains very valuable information, but also covers a lot of different aspects of the paper. Perhaps subheadings may help guide the reader through these different parts.

Thank you, Silvan Licher

Reviewer #3: Review

This is an interesting paper, but I feel the analysis could do with expanding (in terms of description, at least). I am also concerned by the fact that UKBB includes people from England, Wales and Scotland (but only those near the assessment centres), whereas SAIL includes participants from across Wales.

Main comments

1. The comparison is between UK Biobank and a Welsh representative sample - so it is possible that differences between the rest of the UK and Wales could account for some/all of the differences observed here.

2. There were only two UKBB assessment centres in Wales, in Swansea and Cardiff. It would be useful to consider restricting the SAIL analysis to participants registered in Swansea or Cardiff, to ensure comparability.

3. "LTCs were identified from Read codes occurring prior to the baseline assessment, using the criteria described by Barnett et al." It would be useful to have more detail here, e.g. how long before the baseline assessment could the Read code occur? (there is presumably a cut-off?)? Was the % of participants with GP records going back 1,2, 5 however-many years the same across UKBB and SAIL? Any differences in READ recording between UK and UKBB? Could the list of READ codes be provided in supplementary material?

4. Similarly, "Finally, we identified combinations of three LTCs from previous studies of multimorbidity clusters,7,12 and assessed all possible combinations of the conditions within each cluster." - more information both in main text and as supplementary would be useful - I could not understand from this description what the outcomes were.

5. For outcomes, what was the cut-off date in each dataset (i.e. censoring date)? Is length of follow-up comparable?

6. I found the description of the standardisation a little confusing. I am used to standardisation being either direct or indirectly standardised rates, i.e. here would have direct standardisation within strata of age/sex (with age grouped) and then within strata of age/sex/SES, comparing the number of cases (multimorbidity rates) in UKBB with the number expected from SAIL rates. I think something similar is done here, except that a parametric model is used rather than strata and direct comparison of numbers of rates. More detail on how this model was derived would be helpful - i.e. exactly how were fractional polynomials used, what degree did you go up to, how did you assess model fit, etc. Did the negative binomial models fit well? Were any other models considered?

The fit of the models in SAIL is crucial to the interpretation of their fit to the UKBB data.

How did you estimate the confidence intervals for the UKBB estimates?

7. There is a similar lack of detail for the survival models. Weibull were found to fit best, but what other options were considered? What criteria were used to arrive at the best-fitting model? This information is unclear: "censoring participants dying of non-cardiovascular causes and coding as event status '0'". Was there an overall censoring date, and did it differ between participants (and was length of follow-up similar for SAIL and UKBB)?

8. Why was Poisson used for unscheduled hospitalisation? What (exactly) was the outcome? Why was time to first unscheduled hospitalisation not used? Were there many people with multiple unscheduled hospitalisations and how were they dealt with? Was the assumption made that 1 person with 5 unscheduled hospitalisations over 5 years contributes the same information as 5 people each with 1 unscheduled hospitalisation over 1 year? I would think it was possible that there might be over-dispersion, or that some people may be prone to several unscheduled hospitalisations?

9. Results - figure 1 -I would actually like to see a table for both figure 1 and 2 with observed and expected numbers in it, rates and their CIs. It is hard to compare rates on the figure when the prevalences are quite low.

10. Results - "After additionally standardizing by socioeconomic deprivation (red circles), however, the expected LTC counts in SAIL". Are these the expected LTC counts in SAIL, or the expected counts in UKBB given the rates in SAIL? I had assumed the latter, given the description of the methods. Thus on Figure 2, I would expect to see observed UKBB counts, and UKBB expected given SAIL age/sex model and UKBB age/sex distribution, and UKBB expected given SAIL age/sex/SES model and UKBB age/sex/SES distribution. To me "SAIL age/sex standardised to UKBB" implies predicted SAIL counts given UKBB age/sex model and SAIL age/sex distribution.

11. I would like to see confidence intervals on all figures. Would be useful also to see obs/expected (as in the more usual standardisation), especially for the higher counts of multimorbidity as the prevalences are so low it is hard to compare them on the figure. 

12. What was the age/sex/SES distribution of UKBB and SAIL? I would expect to see this in a table in the main paper.

13. In figure 3a, is the absolute mortality risk shown for a particular age/SES group, or averaged over the population as a whole, or something else? I can't relate the text about a specific age group to the figure?

14. For figure 3b, would be better to have points not blocks and then show the confidence intervals for both SAIL and UKBB. The same for figure 4b and 5b. And figure 6.

15. This sentence "Importantly, the rank-order of LTCs in terms of their mortality risk was broadly similar between the two datasets." - what does this mean? How do you judge "broadly similar"? If this is important, I'd expect to see a table/figure showing the rank-order of LTCs in terms of mortality in the two datasets and some sort of comparison. If this is about the HRs shown in Figure 6, then having figure 6 ordered either by SAIL or UKBB HR would help (am not sure what it is ordered by, is roughly mortality risk but doesn't seem to be exactly HR in either dataset?).

16. Discussion - "than in a similarly aged representative UK sample from SAIL databank," - I thought the SAIL sample was only Welsh?

17. "Moreover, although generally less prevalent in UK Biobank, many LTCs had a similar relative risk of mortality between the two datasets.". This is a bit vague. How many? How do you judge "similar"?

18. "However, UK Biobank is likely to underestimate the impact of multimorbidity in people with greater numbers of LTCs (e.g. 4 or more) ". I think care is needed here - in all cases the comparison is to having no LTCs - whereas the sentence implies that the comparison is limited to those having 4 or more LTCs. The issue is that the difference between predicted impact of LTCs in SAIl and UKBB increases as the number of LTCs increases.

19. Was ethnicity examined at all here? Was ethnicity similarly distributed across SAIL and UKBB?

20. "SAIL includes data from Wales only, whereas UK Biobank comprises England, Scotland and Wales, although this is unlikely to have a substantial impact on our findings". This feels like quite a big claim to make with no evidence - you could compare the SAIL results to UKBB both only including people from Cardiff and Swansea - although numbers will be smaller, it would help to see how much the differences are due to Wales vs England vs Scotland?

[LINK]

---

## [Decision Letter · Decision Letter 1]

16 Nov 2021

Dear Dr. Hanlon,

Thank you very much for re-submitting your manuscript "Associations between multimorbidity and adverse health outcomes in UK Biobank and the SAIL Databank: A comparison of longitudinal cohort studies" (PMEDICINE-D-21-02504R1) for review by PLOS Medicine.

I have discussed the paper with my colleagues and the academic editor and it was also seen again by three reviewers. I am pleased to say that provided the remaining editorial and production issues are dealt with we are planning to accept the paper for publication in the journal.

[LINK]

We look forward to receiving the revised manuscript by Nov 23 2021 11:59PM.   

Sincerely,

Beryne Odeny (Louise Gaynor-Brook, MBBS PhD)

PLOS Medicine

plosmedicine.org

Requests from Editors:

Comments from Reviewers:

Reviewer #1: We thank the authors for addressing our previous comments, and have no further major issues to report. It may however be considered to further account for the significant difference in prior socioeconomic status distribution between SAIL and UK Biobank from Table 1 (which moreover might be expected to be 20% for each quintile, in a truly representative sample), if possible.

Reviewer #3: 

1. I think the abstract should mention that UKBB is UK-wide (2 centres in Wales) and that the SAIL is Wales.

2. Also can you clarify in the abstract why UKBB has half the expected number of participants? (I think because of linkage, but it would be useful to have this info in the abstract).

3. Abstract - I don't think UKBB accurately estimates risks (because you say absolute risks are underestimated), but relative risk/risk ratio/hazard ratio?

4. Figure 2 could have confidence intervals on the expected number? I would expect CIs to be in the vertical plane not the horizontal. Also it would be useful if the legend for "expected" indicated that this was expected given the model in SAIL.

5. Supplementary Table 2 indicates that once SES is taken into account, there seem to be more people in UKBB with the high number of morbidities, compared to what would be expected from SAIL. The confidence intervals also look very narrow - though I appreciate this could be due to rounding to the nearest whole number? 

6. "We then compared the distribution of LTC count in UK Biobank with SAIL after standardizing SAIL to the age/sex distribution, and then the age/sex/socioeconomic status distribution, of UK Biobank.". I found this description a bit confusing. My reading of the rest of the methods is that you fitted a model in SAIL, then used it to predict outcomes in UK Biobank. I don't see how this means that SAIL was standardized to the age/sex/SES distribution of UK biobank? I would have described it as the expected number in each LTC category if the rates in SAIL applied in UKBB? But this may just be semantics and the description of the methods is clear.

7. Thanks for the sensitivity analyses restricting to the populations of Cardiff and Swansea. For mortality I agree that the main results are unaffected - but for unscheduled hospitalisations, the results appear much closer in the sensitivity analysis,

[LINK]

---

## [Editor Report · Decision Letter 2]

26 Jan 2022

Dear Dr Hanlon, 

On behalf of my colleagues and the Academic Editor, Dr. Sanjay Basu, I am pleased to inform you that we have agreed to publish your manuscript "Associations between multimorbidity and adverse health outcomes in UK Biobank and the SAIL Databank: A comparison of longitudinal cohort studies" (PMEDICINE-D-21-02504R2) in PLOS Medicine.

Before your manuscript can be formally accepted you will need to complete some final editorial requests and formatting changes, which you will receive in a follow up email. Please be aware that it may take several days for you to receive this email; during this time no action is required by you. Once you have received these formatting requests, please note that your manuscript will not be scheduled for publication until you have made the required changes.

PRESS

Sincerely, 

Louise Gaynor-Brook, MBBS PhD 

Associate Editor 

PLOS Medicine